# Exploiting ferrofluidic wetting for miniature soft machines

Mengmeng Sun [1,7], Bo Hao [1,7], Shihao Yang [1], Xin Wang [1], Carmel Majidi [2] ✉ & Li Zhang [1,3,4,5,6] ✉

Miniature magnetic soft machines could significantly impact minimally invasive robotics and biomedical applications. However, most soft machines are limited to solid magnetic materials, whereas further progress also relies on fluidic constructs obtained by reconfiguring liquid magnetic materials, such as ferrofluid. Here we show how harnessing the wettability of ferrofluids allows for controlled reconfigurability and the ability to create versatile soft machines. The ferrofluid droplet exhibits multimodal motions, and a single droplet can be controlled to split into multiple sub-droplets and then re-fuse back on demand. The soft droplet machine can negotiate changing terrains in unstructured environments. In addition, the ferrofluid droplets can be configured as a liquid capsule, enabling cargo delivery; a wireless omnidirectional liquid cilia matrix capable of pumping biofluids; and a wireless liquid skin, allowing multiple types of miniature soft machine construction. This work improves small magnetic soft machines' achievable complexity and boosts their future biomedical applications capabilities.

Amoeba-like soft machines capable of dramatic shape change, splitting, and coalescing could be transformative for real-world applications. Such systems are exciting candidates for biomedical applications such as targeted drug delivery[1,2], minimally invasive surgery[3,4], cell transplantation[5,6], and medical catheter[7,8]. Current efforts towards the development of miniature soft-bodied machines have focused on a variety of actuation mechanisms, such as optical fields[9-11], chemical actuation[12-14], magnetic fields[15-18], where the externally applied magnetic field is a promising option for actuating these machines safely, rapidly, precisely, and dexterously[19-23]. Magnetic small-scale soft machines prepared by adding hard magnetic micron particles to solid soft materials such as polydimethylsiloxane (PDMS), Ecoflex, and hydrogels, are especially promising since they can achieve a wide range of motion patterns, variable morphologies, and multiple functions[24-33]. However, their limited deformability makes it difficult for miniature machines to navigate congregated and narrow spaces,

such as tiny lumens with opening dimensions comparable to or much smaller than the machine. These inherent problems essentially challenge the existing approaches to constructing small-scale soft machines from solid magnetic materials; thus, it is necessary to expand further the materials for building miniature soft-bodied machines and their new functions.

Magnetic liquid materials (e.g., ferrofluid) may provide unique insights and feasible solutions to construct novel multifunctional miniature machines[34,35]. As a magnetic liquid material, ferrofluids are colloidal liquids made of nanoscale ferromagnetic or sub-ferromagnetic particles suspended in a carrier fluid (usually organic solvent or water). Compared to magnetic solid materials, softer, milder ferrofluids can use their extreme deformability to pass through narrow regions with openings smaller than their nominal size and can be reconfigured into small-scale machines with multiple functions. Recent research works have demonstrated that oil-based ferrofluid

[1]Department of Mechanical and Automation Engineering, The Chinese University of Hong Kong, Hong Kong, China. [2]Department of Mechanical Engineering, Carnegie Mellon University, Pittsburgh, PA 15213, USA. [3]Chow Yuk Ho Technology Center for Innovative Medicine, The Chinese University of Hong Kong, Hong Kong, China. [4]Multi-Scale Medical Robotics Center, Hong Kong Science Park, Shatin NT, Hong Kong SAR, China. [5]Department of Surgery, The Chinese University of Hong Kong, Hong Kong, China. [6]CUHK T Stone Robotics Institute, The Chinese University of Hong Kong, Hong Kong, China. [7]These authors contributed equally: Mengmeng Sun, Bo Hao. ✉e-mail: cmajidi@andrew.cmu.edu; lizhang@cuhk.edu.hk

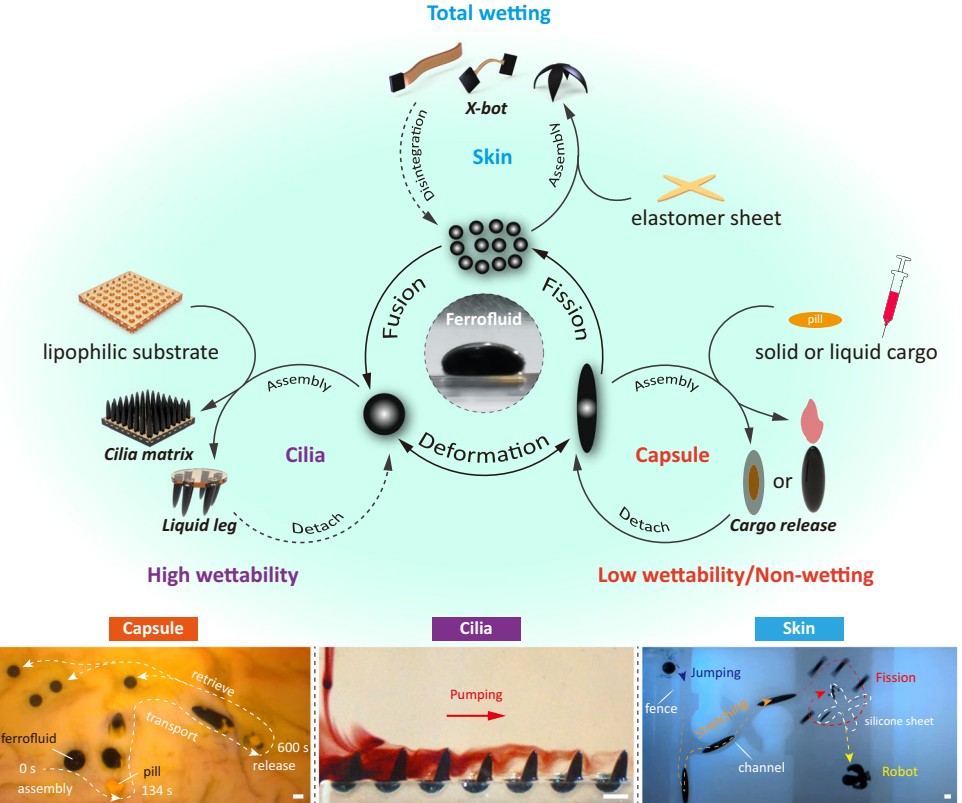

**Fig. 1 | Multifunctionality of wetting ferrofluid droplets.** Schematic and experiment snapshots showing the versatility of ferrofluid droplets: serving as liquid capsules, serving as wireless liquid cilia, serving as wireless liquid skin. Scale bars, 2 mm.

droplets of different sizes act as various tiny machines[36–41]. For example, reconfiguring centimeter-scale ferrofluid droplets as soft machines via an array of electromagnetic coils enables liquid cargo delivery and manipulation of delicate objects[36]. Furthermore, the transport and mixing of liquid samples in lab-on-a-chip applications is realized by driving millimeter-scale ferrofluid droplets using magnetic field gradient forces generated by the electromagnetic navigation floor[37]. Despite many advances in using ferrofluid to construct miniature soft machines, existing machines based on ferrofluid droplets only consider the case at low wettability with the substrate (contact angle $\theta_c$, $90° \leq \theta_c < 180°$), meaning that the interaction strength between solid-liquid is extremely weak[42–46]. However, there are different wetting dynamics between the ferrofluid droplet and the interface[44–51]; in addition to the low wetting case, there is also high wetting and complete wetting, but the application scenarios for different wetting characteristics between ferrofluid droplets and substrates have not been fully explored. Moreover, these ferrofluid droplet-based machines are limited to a few simple modes of motion, such as stretching and rolling, due to their reliance on magnetic field gradient forces for actuation. Furthermore, while bringing better adaptability to these droplet-based machines, the high deformability limits their mechanical properties. The dynamic behavior of millimeter-scale ferrofluid droplets in wetting with different interfaces is underutilized, and the individual dynamics under magnetic torque have not been adequately investigated; thus, it is challenging to realize their full potential to be reconfigured as miniature soft machines.

Here, we demonstrate that versatile miniature soft machines can be constructed by exploiting the wetting properties and reconfigurability of ferrofluids (Fig. 1 and Supplementary Fig. 1). First, the wetting dynamics of ferromagnetic fluid droplets at different solid interfaces are studied. When the interaction between ferrofluid and substrate is weak (low wettability with substrate), the magnetic torque generated by the spatiotemporally programmed magnetic field drives ferrofluid

droplets to perform stretching, jumping, rotating, tumbling, kayaking, and wobbling motions. Due to their liquid properties, the ferrofluid droplets can split and fuse along the line or the plane in a controlled manner, and the number of fission is highly controlled. Compared with the ferrofluid droplets driven by the magnetic field gradient force[36,37,41], the magnetic torque-driven ferrofluid droplets have various motion modes and split modes. The ferrofluid droplets are highly adaptable to both artificial and biological environments by exploiting multimodal motion and controllable fission-fusion properties. The ferrofluid droplets can cross over successive obstacles, upstairs and through designated holes in walls in a jumping mode, navigate through highly curved small gaps and sharp turns in a stretching pattern, and pass through narrow comb-like channels using fission and fusion. Moreover, the ferrofluid droplets can be reconfigured into miniature machines with multiple functions using wetting dynamics. At low wettability with cargo, the droplets can be reconfigured to serve as liquid capsules for transporting liquid or solid cargo, which can travel through tortuous narrow channels to reach targeted positions and release the load-on-demand. However, it is hard for the solid capsule to traverse cavities when its size is comparable with the cross-sectional dimension of these confined spaces. When the interaction between ferrofluid droplets and the interface is strong (high wettability, $0° < \theta_c < 90°$), the controllable deformability of the droplets allows them to act as arrays of liquid cilia, which are programmed to pump biological fluids. Conventional artificial cilia are in a solid state[18,27], and their morphology cannot be easily changed, so they maintain a rigid structure without a magnetic field, which may affect fluid flow. Without an applied magnetic field, our liquid cilia will shrink to the bottom of the substrate and take on a spherical shape, thus reducing the impact on fluid flow. When the interaction between ferrofluid droplets and the interface is very strong (total wetting, $\theta_c = 0°$), the droplets can be reconfigured to serve as an active wireless liquid skin, which can controllably navigate near inanimate targets and then transform it into

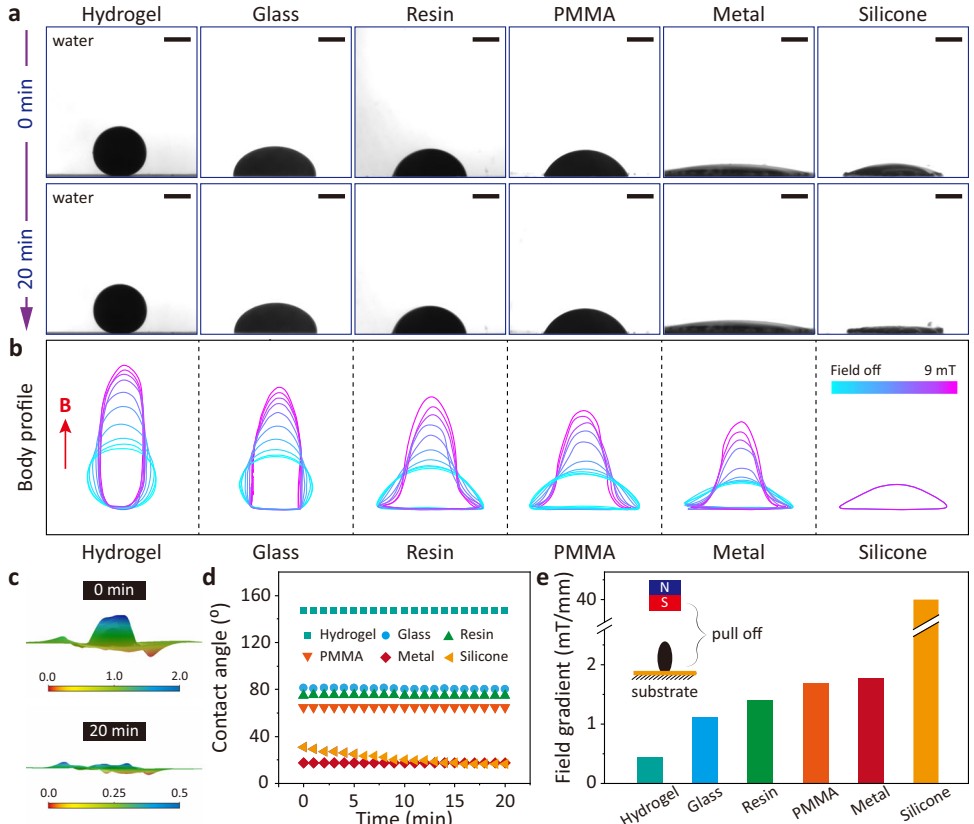

**Fig. 2 | Wetting dynamics of the ferrofluid droplet. a** Images of ferrofluid droplets (2 µL) on different substrates (hydrogel, glass, resin, polymethyl methacrylate (PMMA), metal (copper), silicone) at 0 min and 20 min after the dripping. **b** The overlapped body profiles of ferrofluid droplets on different substrates as the uniform magnetic field strength is increased from magnetic field off to 9 mT. The red arrow indicates the direction of the magnetic field. Color bar represents normalized magnetic field strength. **c** The morphology of a ferrofluid droplet at different moments when it is added to a silicone substrate. Color bars represent normalized heights. **d** Time evolution of the contact angle of ferrofluid droplets on different substrates. See Supplementary Data File 1 for source data for the graph. **e** The quantitative pulling off strength of ferrofluid droplets to different surfaces. See Supplementary Data File 2 for source data for the graph. All the suspending phase is water. Scale bars, 1 mm.

a soft machine through an adhesive strategy. Unlike the passive magnetic spray[28], our ferrofluid droplet has multimodal motion capabilities that act as active, movable skin. Our proposed method to construct soft-bodied machines via the wettability of ferrofluid can inspire new construction strategies and achieve various unprecedented functionalities that could find broad applications in biomedical engineering.

## Results and discussion
### Wetting dynamics of ferrofluid droplets
We use an oil-based colloidal ferrofluid composed of nanoscale $Fe_3O_4$ particles suspended in a carrier fluid (fluorocarbon oil). Each magnetic particle is thoroughly coated with a surfactant to inhibit clumping. To understand and control ferrofluid droplets, it is necessary first to understand the wetting dynamics of ferrofluid droplets with different substrates, which can be quantified by the contact angle. We have characterized the wettability of ferrofluid droplets on different substrates by measuring the contact angle of the ferrofluid droplet as a function of time without applying a magnetic field. As shown in Fig. 2a, without external magnetic field, the ferrofluid droplets do not adhere to the hydrogel substrate (low wettability) and can stick to the glass, resin, polymethyl methacrylate (PMMA), and metal substrates (high wettability). The contact angle of ferrofluid droplets on hydrogel, glass, resin, PMMA, and metal substrates remained essentially constant from 0 to 20 min. The contact angle is different from hydrocarbon oil with various substrates in the air environment (Supplementary Fig. 2). However, the contact angle of the ferrofluid droplet on the silicone surface changes significantly. This change is mainly since the liquid

carrier medium of the ferrofluid is absorbed by the silicone substrate, leaving only a magnetic nanoparticle film on the surface[38,45]. Furthermore, with the external magnetic field, the morphology of the ferrofluid droplets on the different substrates changes as the magnetic field strength varies from 0 to 9 mT (Fig. 2b). Only the ferrofluid droplets on the silicone surface are excluded, as the surface magnetic nanoparticle film does not change at lower magnetic field strengths. The morphological changes of the ferrofluid droplets on the silicone substrate are characterized by 3D optical microscopy (Fig. 2c). The maximum height of ferrofluid droplets in the initial state is approximately 2 mm. After 20 min, the maximum height of ferrofluid droplets on the surface of the silicone substrate is about 0.5 mm. As shown in Fig. 2d, quantitative data on the contact angle on different substrates shows that the contact angle of the ferrofluid on the hydrogel substrate is the largest at approximately 147°. The contact angles on glass, resin, and PMMA substrates are similar, at 80°, 75°, and 65°, respectively. In the initial state, the ferrofluid has a contact angle of 30° on the silicone surface, but after 20 min, it has a consistent contact angle of approximately 16° with the metal surface. The adhesion capacity of the ferrofluid on different substrates is characterized by constructing magnetic fields to test the magnetic field strength required to detach the droplets of ferrofluid on different substrates (Fig. 2e). Since the ferrofluid droplets on the silicone surface have formed a hard layer of magnetic nanoparticles, a magnetic field strength of 300 mT is required to detach them. In addition, when the silicone substrate with ferrofluid deposited is submerged in isopropanol (IPA) solution, the rigid layer of magnetic nanoparticles adhering to the robot surface is destroyed by

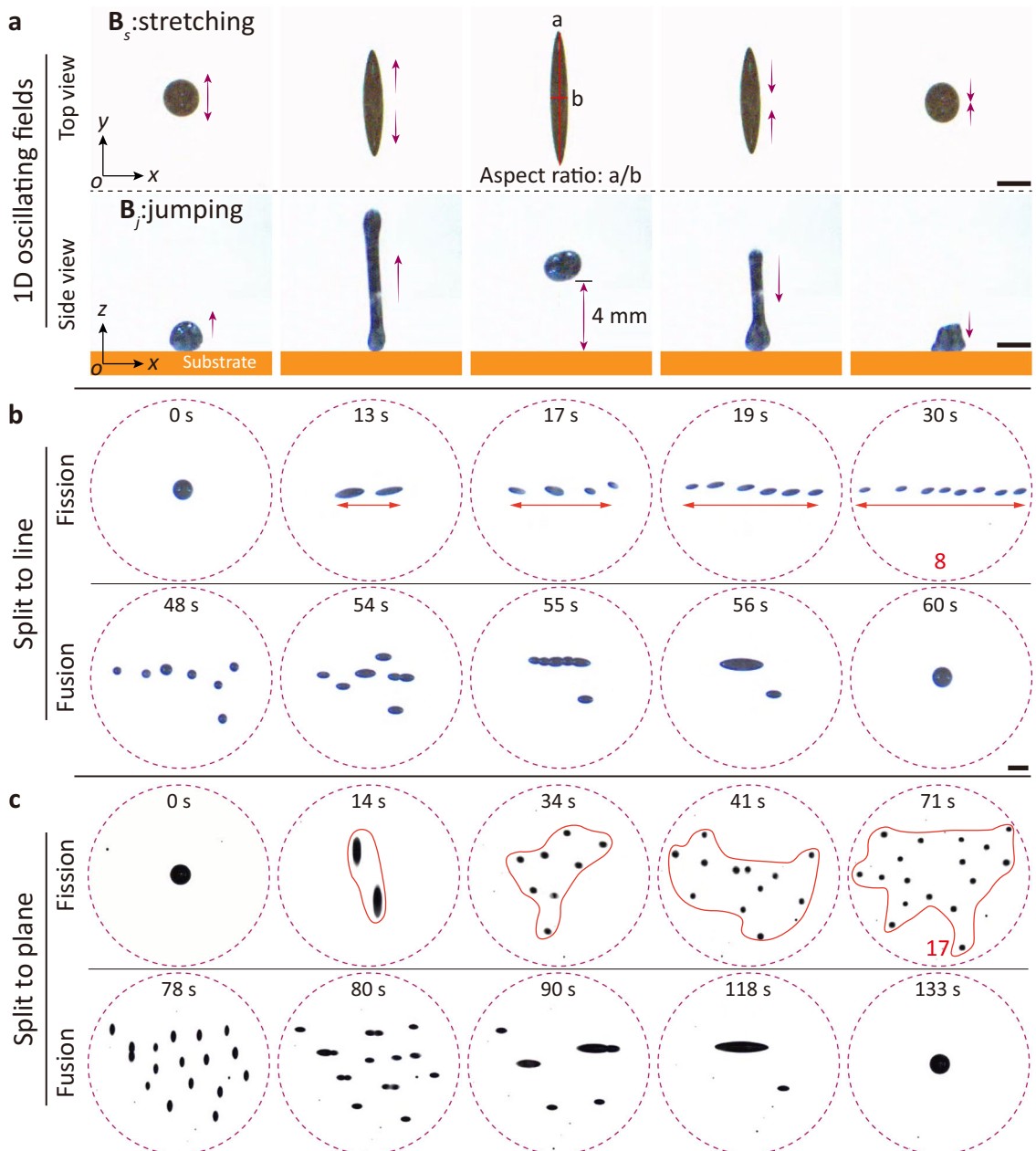

**Fig. 3 | Fission and fusion scenarios. a** Stretching and jumping. a and b represent the length and width, respectively. Ferrofluid droplets exhibiting stretching and jumping postures under 1D oscillating fields $\mathbf{B}_s(t)$ and $\mathbf{B}_j(t)$, respectively. $\mathbf{B}_s(t)$: frequency $f = 1$ Hz and magnitude $B_m = 9$ mT. $\mathbf{B}_j(t)$: $f = 100$ Hz and $B_m = 9$ mT. **b** Under a 2D oscillating magnetic field $\mathbf{B}_{Osc}(t)$, one ferrofluid droplet splits into eight tiny droplets along a line and subsequently fuses. $\mathbf{B}_{Osc}(t)$: $f = 25$ Hz and $A_O = C_O = 9$ mT. **c** Under a 3D wavy magnetic field $\mathbf{B}_{Wav}(t)$, one ferrofluid droplet splits into seventeen microdroplets onto the plane and subsequently fuses. The yellow rectangle represents the substrate in the side view. All the substrates are made of hydrogel. Scale bars, 2 mm.

stress concentration arising from a mismatch between the expansion ratio of the stiff layer and the silicone elastomer sheet. The magnetic nanoparticles are gradually disintegrated and enriched from the silicone elastomer sheet under a magnetic field (Supplementary Fig. 3). The hydrophobicity of the matrix mainly determines the wetting properties of ferrofluid droplets. Therefore, different hydrophobic substrates can be selected to construct miniature soft robots according to the needs.

## Individual ferrofluid droplet dynamics

By varying the magnitude and direction of the external magnetic field, the ferrofluid droplets exhibit different shape-changing mechanisms. When ferrofluid droplets do not adhere to the substrate, which is made of hydrogel (low wettability), changing the spatial configuration of the external magnetic field enables locomotion modalities like stretching, jumping, rotating, tumbling, kayaking, and wobbling (Supplementary Movie 1, the spatial distribution states of magnetic fields plotted by the MATLAB R2017a can be queried in Supplementary Fig. 4). When a one-dimensional (1D) oscillating magnetic field is applied, the ferrofluid droplet exhibits two modes of motion: stretching and jumping. As shown in Fig. 3a, a ferrofluid droplet with an initial diameter of 2 mm stretches back and forth in situ in the $x$-$y$ plane when a 1D horizontally oscillating magnetic field $\mathbf{B}_s(t) = B_m\left[\sin(2\pi f t)\mathbf{e}_y\right]$ is applied; here $B_m$ is the magnitude of magnetic field, $f$ is the frequency of magnetic field, $\mathbf{e}_y$ is the unit vector along the $y$ axis (hereafter, $\mathbf{e}_x$ and $\mathbf{e}_z$ is that along the $x$ and $z$ axes, respectively). The elongation depends on the strength of the external magnetic field, the maximum length is 7.5 mm at 9 mT,

and the corresponding aspect ratio is 9 (Supplementary Fig. 5). In addition, if the angle between the ferrofluid and the substrate (defined as the pitch angle) is increased, the ferrofluid droplet could induce translational motion while performing a reciprocal stretching pattern, which is caused by the friction from the substrate (Supplementary Fig. 6). Supplementary Figure 7 demonstrates the velocity of the ferrofluid droplet with the stretching mode as a function of the frequency of external magnetic field. When a 1D vertically oscillating magnetic field $\mathbf{B}_j(t) = B_m[\sin(2\pi ft)\mathbf{e}_z]$ is applied, the ferrofluid droplet still displays a stretching motion at low frequency ($f < 50$ Hz), the maximum aspect ratio of the droplet is 5.5 (Supplementary Fig. 8). If the frequency of the magnetic field is increased ($f > 50$ Hz), the ferrofluid exhibits a jumping behavior in situ (Fig. 3a). The mechanism of ferrofluid droplet jumping is that the high-frequency alternating magnetic field magnetizes the droplet, which exhibits N-S poles and then reverses the direction of the alternating magnetic field. As the polarization of droplets is unable to respond in time, the external magnetic field exerts a repulsive force on the droplet, causing it to jump up (Supplementary Fig. 9). The jumping height of the ferrofluid droplet is related to the magnitude of the externally applied magnetic field. When the frequency is higher than 50 Hz, the greater the magnitude of the magnetic field, the higher the droplet jump height, the maximum jump height is 4 mm ($f = 100$ Hz, $B_m = 9$ mT). By changing the pitch angle of the magnetic field, the ferrofluid droplets can also achieve directional jumping (Supplementary Fig. 10). When a two-dimensional (2D) rotating magnetic field is applied, the ferrofluid droplet exhibits two motion modes: rotating and tumbling $\mathbf{B}_r(t) = B_m[\cos(2\pi ft)\mathbf{e}_x - \sin(2\pi ft)\mathbf{e}_y]$, $\mathbf{B}_t(t) = B_m[\cos(2\pi ft)\mathbf{e}_x - \sin(2\pi ft)\mathbf{e}_z]$, Supplementary Figs. 11a, 12). When a three-dimensional (3D) conical magnetic field is applied, the ferrofluid droplet exhibits two modes of motion: kayaking and wobbling $\mathbf{B}_k(t) = B_y\mathbf{e}_y + B_m[\cos(2\pi ft)\mathbf{e}_z - \sin(2\pi ft)\mathbf{e}_x]$, $\mathbf{B}_w(t) = B_z\mathbf{e}_z + B_m[\cos(2\pi ft)\mathbf{e}_x - \sin(2\pi ft)\mathbf{e}_y]$, Supplementary Figs. 11b and 13). Compared with rigid robots and elastomer-based soft robots, ferrofluid droplets possess higher degrees of freedom. They can achieve high maneuverability using multimodal motion, thus negotiating obstacles and textures in complex environments.

Ferrofluid droplets have both magnetic responsiveness and fluidity. Excellent magnetic responsiveness endows the ferrofluid droplets with multiple modes of mobility, and the fluidity of the liquid allows the ferrofluid to perform extreme deformations, such as fission and fusion, and the process is reversible. We adjust the ratio between magnetic torque and drag torque to trigger fission and fusion processes (Supplementary Movie 2). As shown in Fig. 3b, the ferrofluid droplet with an initial diameter of 2 mm first splits along the line into 8 sub-droplets (with an average diameter about 1 mm) and then gradually merges into the initial state, driven by the 2D oscillating magnetic field $\mathbf{B}_{Osc}(t)$ (the spatial distribution states of magnetic fields can be queried in Supplementary Fig. 14), which consists of a constant component and a sinusoidal component perpendicular to the constant component:

$$\mathbf{B}_{Osc}(t) = [A_O \sin(2\pi ft)\cos\theta + C_O \sin\theta]\mathbf{e}_x + [A_O \sin(2\pi ft)\sin\theta + C_O \cos\theta]\mathbf{e}_y \quad (1)$$

where $A_O$ is the amplitude of the sinusoidal signal, $C_O$ is the intensity of the constant component, and $\theta$ is the angle between the constant component and the $y$-axis. The oscillating magnetic field transforms the spherical ferrofluid droplet into the shuttle shape, and the shuttle-shaped ferrofluid starts to oscillate under the magnetic torque. As the frequency increases, the viscous resistance from water begins to increase, so its surface tension cannot maintain the integrity, and the whole droplet begins to split. Their alignment direction is controlled

by the direction of the oscillating magnetic field. As the frequency decreases, the magnetic dipolar force on the ferrofluid sub-droplet dominates its dynamic behavior. The ferrofluid droplets begin to approach each other due to the magnetic attraction. As the sub-droplets proceed into close contact, their interfaces flatten, and the fluid film between the droplets drains. Eventually, when the fluid film becomes thin enough, the van der Waals intermolecular forces dominate and cause the liquid film to rupture, ultimately contributing to droplet coalescence. To achieve further splitting of millimeter ferrofluid droplets, one strategy is to increase the distance between sub-droplets by inducing repulsive magnetic forces between them, weakening their attractive interactions and preventing them from reassembling. To achieve the expected fission and fusion scenario, a 3D wavy magnetic field $\mathbf{B}_{Wav}(t)$ is designed (the spatial distribution states of magnetic fields can be queried in Supplementary Fig. 14), which is composed of a 1D oscillating and a 2D rotating magnetic fields:

$$\mathbf{B}_{Wav}(t) = B_m \cos(\alpha(t))[\sin(2\pi f_1 t)\mathbf{e}_z] + B_m \sin(\alpha(t))[\cos(2\pi f_2 t)\mathbf{e}_x - \sin(2\pi f_2 t)\mathbf{e}_y] \quad (2)$$

where $B_m$ is the maximum field strength, $\alpha$ is the minimal precession angle, $f_1$ is the frequency of 1D oscillating magnetic field, $f_2$ is the frequency of the 2D rotating magnetic field. During actuation, the time-dependent precession angle $\alpha(t)$ is changing between $\alpha_0$ and $\pi - \alpha_0$. As shown in Fig. 3c, driven by the 3D wavy magnetic field $\mathbf{B}_{Wav}(t)$, a 2 mm diameter ferrofluid droplet splits into 17 sub-droplets (with an average diameter of about 400 $\mu$m), and spread to the plane and then gradually fuse to the initial state when decreasing the input frequency. In addition, the controllable splitting of ferrofluid droplets is achieved by modulating the magnetic field distribution (Supplementary Fig. 15). This scale-reconfiguration property enables ferrofluid droplets to travel freely through complex environments with highly variable feature sizes.

## Environmental adaptability

It remains challenging for miniature robots to navigate freely in complex environments with obstacles, textures, and variations. To tackle this challenge, when the ferrofluid is at low wetting with the surrounding environment, a combination of multimodal motion, deformation, and scale reconfiguration capabilities e7nables controlled locomotion of ferrofluid droplets in complex artificial and biological environments, such as continuous fences, wide gaps, uneven bladder lining, folded stomach lining, narrow lumen (Supplementary Movie 3 and Supplementary Movie 4). As illustrated in Fig. 4a, we design a continuous array of three fences with peak-to-peak distances of 6 mm and heights of 2.2 mm, 3.6 mm, and 4.5 mm, respectively. Experimental results demonstrate that the ferrofluid droplet successively overrides these fences in the jumping mode, typically consisting of four stages: elongation, contraction, descent, and landing. Firstly, a directional 1D oscillating magnetic field $\mathbf{B}_s(t)$ ($f = 100$ Hz, $B_m = 9$ mT, angle of 80° to the $x$-axis) is applied and then the ferrofluid elongates along the 80° direction. Secondly, the ferrofluid droplet starts to contract itself and reaches its highest point. Then, the droplet starts to descent under the force of gravity. Eventually, the ferrofluid droplet crosses the fence and lands at the target end. In addition, the droplet can jump over three successive steps to reach the highest point, each at a height of 2 mm, even when facing a specified hole in a wall with an inner diameter of 8 mm and a height of 2 mm above the floor (Fig. 4b and c). The capability to maneuver in the confined space is crucial for medical robots operating in a lumen or within cavities inside the human body, so the adaptive deformation behavior of the ferrofluid droplet in the confined space is investigated after. The ferrofluid droplets can passively generate large deformations, using the stretching mode to pass ring-

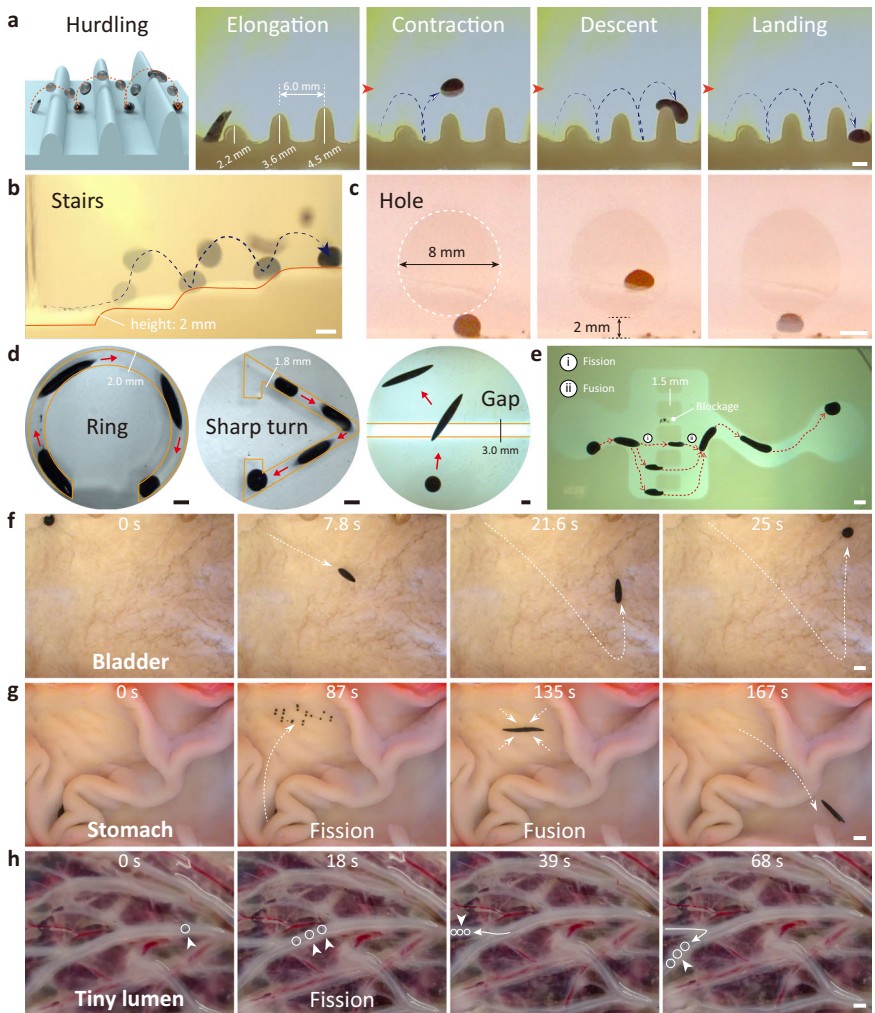

**Fig. 4 | Multimodal locomotion, deformation and scale-reconfiguration over the complex environment. a** Directional hurdling. The ferrofluid droplet jumps over successive obstacles under a 1D oscillating magnetic field (100 Hz, 9 mT). **b** Jumping up the stairs. **c** Jumping over designated holes in the wall. **d** The ferrofluid droplets adaptively deform and actively pass through a circularly curved channel, a sharp turn and a gap. **e** The ferrofluid droplet utilizes splitting behavior and passes through the comb-constrained channel in a stretching motion mode. **f** The ferrofluid droplet tumbles over the uneven bladder lining. **g** The demonstration of a ferrofluid droplet maneuvering on the folded stomach lining. **h** Ferrofluid droplet navigates within the unstructured narrow lumen. All the artificial structures are made of hydrogel. Scale bars, 2 mm.

shaped channels and sharp turns (Fig. 4d). When moving inside an annular channel with an inner diameter of 2 mm, the ferrofluid droplets are constantly turning as they stretch and contract to achieve locomotion by utilizing the supporting force of the sidewall (Supplementary Fig. 16). In a sharp turn with an inner diameter of 1.8 mm, the droplet can also passively adapt to the shape of the turn and creep through it using the stretching mode. The ferrofluid droplet also actively yields deformation to span the gap with a width of 3 mm using the kayaking motion pattern. In addition, we demonstrate that ferrofluid droplet uses controlled fission and fusion to pass through comb-like narrow channels with an inner diameter of 1.5 mm (Fig. 4e). A ferrofluid droplet with a diameter of 3 mm approaches the branch entrance in a stretching mode, but due to the wall effect of the entrance, it is difficult for the ferrofluid droplet driven by magnetic torque to deform directly through the narrow channel. Therefore, a 2D oscillating magnetic field $\mathbf{B}_{Osc}(t)$ ($f = 20$ Hz, $A_O = C_O = 9$ mT) is subsequently applied to allow the ferrofluid droplet to split controllably into three sub-droplets. Then, a 1D oscillating magnetic field $\mathbf{B}_s(t)$ ($f = 10$ Hz, $B_m = 9$ mT) is applied to actuate three sub-droplets to pass through the channels. Finally, these three sub-droplets fuse and then maneuver to the target location.

To investigate the potential of ferrofluid droplets for medical applications and demonstrate their adaptability to the unstructured environment of the human body, we have prepared three typical biological environments with uneven bladder lining, folded stomach lining, and narrow placental vessels. In addition, in vitro cytotoxicity tests have shown that ferrofluid droplets are biocompatible (Supplementary Fig. 17). As demonstrated in Fig. 4f, the inner wall of the porcine bladder is uneven and covered with continuous textured structures, and a ferrofluid droplet with a diameter of 2 mm moves in a controlled tumbling motion mode on the inner wall surface. Due to the unevenness of the inner wall, the ferrofluid droplet slips during navigation, but the overall motion direction is controlled. However, the inner wall of the pig stomach, which is covered with large folds, is a challenge for the miniature robots to perform directional locomotion. The height of the pig stomach folds is about 10 mm, and there are many narrow traps (Fig. 4g). At $t = 0$ s, the ferrofluid droplet is trapped into the recessed area and then slides in situ because of the smooth inner wall under the rotating magnetic field $\mathbf{B}_t(t)$. The ferrofluid droplet is then actuated to break into multiple sub-droplets, crossing the trap ($t = 87$ s). A 2D rotating magnetic field $\mathbf{B}_t(t)$ is applied to induce these sub-droplets to merge into a whole ($t = 135$ s). Hereafter, by increasing the pitch angle of the

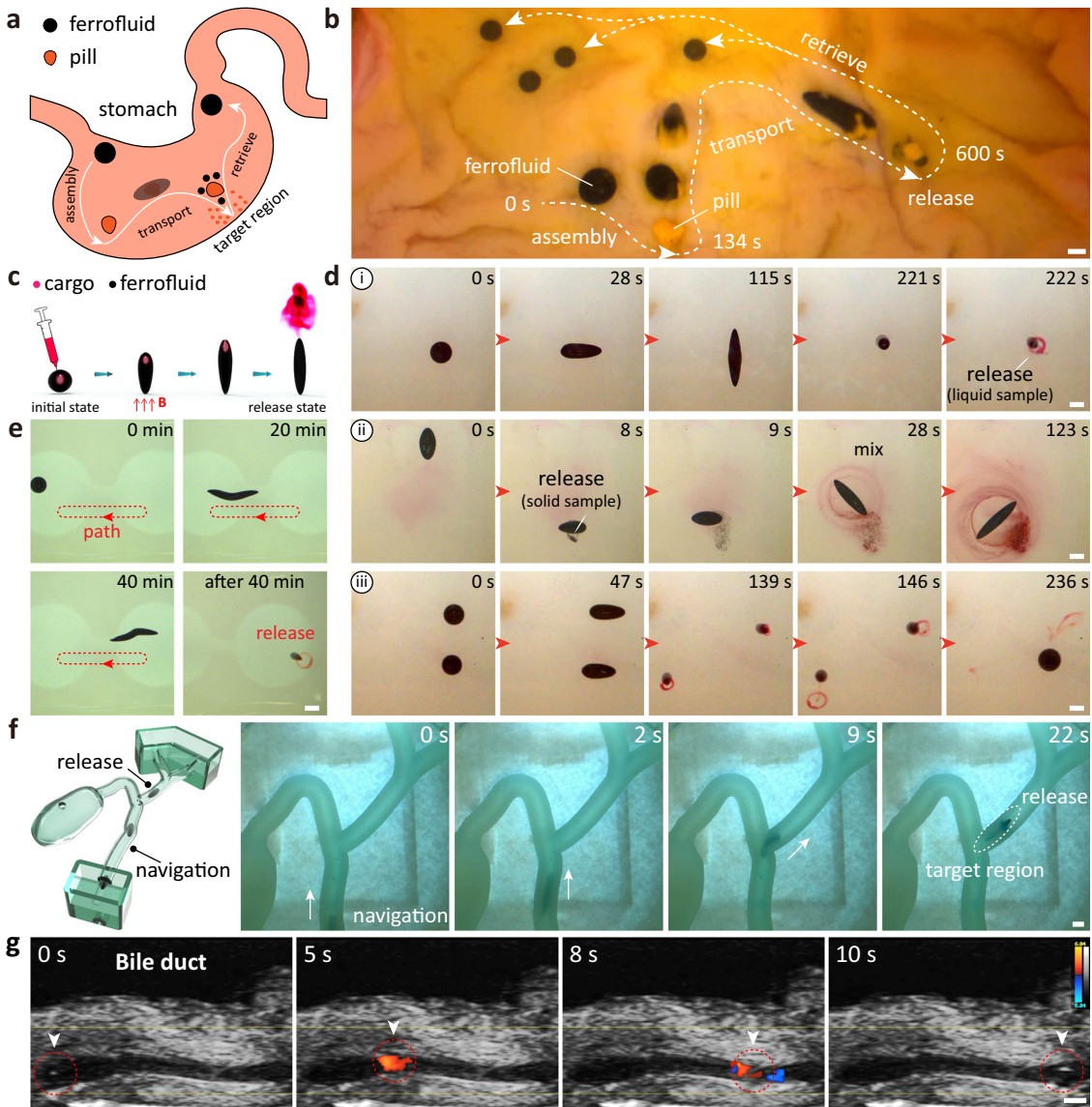

**Fig. 5 | Controllable liquid capsule for on-demand cargo delivery. a** Schematic diagram showing a ferrofluid droplet transforming a traditional pill into an active drug delivery system. **b** The active construction, controlled locomotion and on-demand release of a liquid capsule in an ex vivo animal stomach. **c** Schematic diagram of a liquid capsule that ejects its internal cargo on demand by applying a magnetic field. **d** Liquid capsules maneuver in various motion modes while carrying the liquid or solid cargo and release on demand. (i) Liquid cargo delivery. (ii) Solid cargo delivery and mixing. (iii) Two liquid capsules are actuated together to deliver the liquid cargo and fuse. The substrate is made of hydrogel. **e** The liquid capsule shuttles back and forth in the channel, which is made of hydrogel, for 40 min before ejecting the liquid cargo. **f** The liquid capsule moves in a bile duct phantom made of soft gel, inside whose narrow branches are in stretching mode and release the liquid cargo on demand. **g** Ultrasound Doppler-guided locomotion of the liquid capsule in an ex vivo animal bile duct. Color bar represents the Doppler signal. Scale bars, 2 mm.

rotating magnetic field, the droplets become shuttle-shaped and perform a tumbling motion so that they can cross the folds of the stomach and finally reach the target position ($t$ = 167 s). The ferrofluid droplet with the scale-reconfiguration property also displays high adaptability for the unstructured lumen with variable inner diameters (human placental vessels). As illustrated in Fig. 4h, a ferrofluid droplet with a diameter of 1 mm undergoes rotational-translational motion inside the main vessel with an internal diameter of approximately 1.5 mm ($t$ = 0 s). In order to enter the branches of the lumen with an inner diameter of about 0.8 mm, the ferrofluid droplet starts to split into three sub-droplets with a diameter of about 0.6 mm. At $t$ = 39 s, these three sub-droplets enter the branch lumen smoothly. After 29 s, these three sub-droplets return to the main lumen. Experimental results show that when the ferrofluid is at low wetting with the surrounding environment, by combining multimodal motion, deformation, and scale reconfiguration capabilities, the ferrofluid droplet can navigate freely in complex environments with highly variable feature sizes, which is expected to be applied as wireless soft medical robots in various restricted areas of the human body.

### Miniature liquid capsule at low wettability or non-wetting

The ferrofluid droplet can act as a liquid capsule, carrying the liquid or solid cargo while maneuvering through the complex environment, such as narrow and tortuous ducts, to attain inaccessible target locations, releasing the load on demand. There are two strategies for constructing liquid capsules from ferrofluid droplets; the first transforms passive pills into liquid capsules through an active adhesion strategy; the second constructs liquid capsules by injecting drugs into the inside of ferrofluid droplets using syringe injection. In addition, the

miniature liquid capsule undergoes controlled navigation within the bile duct phantom and the bile duct under the real-time guidance of X-ray fluoroscopy and ultrasound imaging, respectively. As shown in Fig. 5a, b, ferrofluid droplets (4 mm) can actively adhere to the pill, endow it with mobility, and move controllably inside an isolated animal stomach model. When the liquid capsule reaches the infected area, the pills are released on demand by splitting, rather than being scattered randomly throughout the organ. The schematic diagram of loading and discharging the cargo of the liquid capsule via the second strategy is shown in Fig. 5c. First, we inject water-based liquid or solid cargo (placed in a saturated sodium chloride solution) into the ferrofluid droplet. Due to the immiscibility of water and oil phases, the water-based payloads exist stably inside the liquid capsule under the surface tension. But the injection of liquid cargo requires complete wetting between the syringe needle and the ferrofluid droplet (Supplementary Fig. 18). To avoid instability and ejection, a typical 5 µL droplet is injected with a liquid load of approximately 0.5 µL. Then a vertical oscillating magnetic field is applied, which causes the liquid capsule to contract, increasing internal pressure, and the cargo is ejected from the capsule like cuttlefish ejecting ink. As shown in Fig. 5d, the liquid capsule is loaded with a liquid cargo (edible dyes are used here) and maintains stability without rupture in the stretching and tumbling motion pattern, and is finally released by a 1D vertical oscillating magnetic field (Supplementary Movie 5). The mechanisms of drug release and mechanisms to maintain non-leakage during motion are shown in Supplementary Fig. 19. Supplementary Fig. 20 displays a side view of a liquid capsule releasing the liquid cargo process. In addition to delivering liquid cargo, solid edible dyes are placed in a saturated sodium chloride solution and injected into a liquid capsule. The liquid capsule maneuvers to the target position in a tumbling mode and finally releases the solid cargo ($t = 9$ s). Then, the ferrofluid droplet acts as a stirring bar in a rotating manner, thus accelerating the dissolution of the solid food dye (Fig. 5d and Supplementary Movie 5). After 104 s, most of the solid cargo has dissolved, and the fluid field distribution caused by the rotation of the ferrofluid droplet is shown in Supplementary Fig. 21. Additionally, two liquid capsules are driven simultaneously for cargo delivery, and two capsules fuse for retrieval after releasing cargo on-demand (Fig. 5d and Supplementary Movie 5); such a liquid capsule team can deliver higher cargo volumes. The stability of the liquid capsules to carry cargo while experiencing large deformations is also studied. The capsules are first filled with red food dye and then shuttle back and forth through narrow passages for 40 min. The liquid capsule is then activated and releases the red food dye. The experimental results qualitatively demonstrate that the liquid cargo does not leak during the locomotion and large deformation (Fig. 5e).

The ferrofluid droplet has large deformability and shape adaptation to the surrounding environment, allowing the liquid capsule to maneuver to the target location carrying the cargo even in a confined and restricted space. As shown in Fig. 5f, a liquid capsule carries cargo in the bile duct phantom for targeted delivery (Supplementary Movie 6). The inner diameter of the bile duct phantom is approximately 3 mm, and the liquid capsule moves forward by expansion and contraction. After 9 s, the liquid capsule reaches a turn with an inner diameter of approximately 2 mm. The liquid capsule adapts to the shape of a sharp turn, then stretches to the target position before applying an oscillating magnetic field to release the carried cargo ($t = 22$ s). Medical imaging is critical for tracking and navigating liquid capsules in vivo, and liquid capsules are navigated to target locations in real-time under X-ray fluoroscopy and ultrasound imaging. X-ray fluoroscopy imaging shows that single or multiple water-based cargo are stably present within the liquid capsule (Supplementary Fig. 22). Constrained by the limitation of the Helmholtz coil system to the X-ray fluoroscopy imaging area, a robotic arm carrying a three-axis solenoid system is integrated with the X-ray fluoroscopy imaging system here to

actuate the liquid capsule, and the whole setup is shown in Supplementary Fig. 23. Under the real-time navigation of X-ray fluoroscopy imaging, the liquid capsule moves along the inner wall of the bile duct phantom driven by the gradient force of the magnetic field (Supplementary Fig. 24 and Supplementary Movie 6). The liquid capsule performs controlled locomotion in the natural bile duct under the real-time guidance of ultrasound imaging (Fig. 5g and Supplementary Movie 6). The softness of the liquid capsule causes minimal damage to the surrounding borders, and the acoustic response of the liquid capsule itself is different from the tissue and is tracked directly (Supplementary Fig. 25). Such liquid capsules carry and manage cargo locally on demand. The edible dye here can be replaced in the future with potential drugs that are expected to be delivered for cancer, inflammation, and other diseases.

## Miniature wireless liquid cilia at high wettability

Biological cilia are widely found on various organisms in nature in thick rods that induce hydrostatic flow by nonreciprocal motion. Inspired by bio-cilia, several artificial cilia systems have been developed as micro-pumps and micro-mixers[18,27]. Unlike previous elastomer-based cilia, the liquid cilia here are developed by exploiting the ferrofluid droplet's programmable deformation and wetting dynamics. The motion mechanism of a single liquid cilium is demonstrated in Fig. 6a. The high wettability of ferrofluid with the resin is first used to anchor them in the semicircular pit. Then a segmented magnetic field $\mathbf{B}_{\text{Cilia}}(t)$ is applied,

$$\mathbf{B}_{\text{Cilia}}(t) = \begin{cases} B_{\text{m}}\left[\cos(\alpha)\mathbf{e}_x - \sin(\alpha)\mathbf{e}_z\right] & \text{if } \alpha < \theta \\ 0 & \text{if } \alpha \geq \theta \end{cases} \quad (3)$$

where $B_{\text{m}}$ is the maximum field strength, $\alpha = 2\pi f t$ is the rotating angle, $f$ is the frequency, $\theta$ is the maximum rotation angle. Therefore the liquid cilium performs three typical motion stages: in the first stage ($t_1$), the liquid cilium elongates along the magnetic field $\mathbf{B}$ and deforms from an elliptical shape to a spiky shape, its elongation length is determined by the amplitude of the magnetic field; in the second stage ($t_1$–$t_2$–$t_3$), the spiked liquid cilium is rotating in the clockwise direction, and the magnetic field direction determines its rotation direction; in the third stage ($t_3$), the magnetic field is turned off, and the liquid cilium starts to contract from the spiked shape into an ellipse, and the magnetic field determines the magnitude of its turning angle; then the above process is repeated for controlled oscillation. As shown in Fig. 6b, at $t = 0$ s, the ferrofluid droplet remains elliptical and is confined within a 2 mm semicircular pit in the substrate (Supplementary Movie 7). Under the magnetic field $\mathbf{B}_{\text{Cilia}}(t)$, the elliptical droplet elongates into the spiky liquid cilium along 135° measured by Image J 1.5 ($t = 1$ s). After 0.9 s, the liquid cilium rotates to 45° ($t = 1.9$ s). Then the magnetic field $\mathbf{B}_{\text{Cilia}}(t)$ turns off, and the liquid cilium starts to shrink along 45° and transforms from the spiked state to the original state ($t = 2$ s). Unlike natural cilia or other artificial cilia made from elastomer depending on bucking motion[27], this presented liquid cilia based on elongation and contraction exhibit nonreciprocal motion with periodic power stroke and recovery stroke. In its elongation-rotation-contraction motion cycle, the area where the rotational motion travels is the swiping area (the swiping area is a common metric for quantifying nonreciprocal motion and the net fluid flow-induced). As the swiping area increases, the motion of the liquid cilia becomes more nonreciprocal, resulting in increased fluid flow . Furthermore, the size of the swiping area of the liquid cilia can be controlled by programming the amplitude of the external magnetic field and the angle of rotation.

One-dimensional cilia arrays ($1 \times 7$) and two-dimensional cilia matrix ($9 \times 9$) are created by arranging ferrofluid droplet. As shown in Fig. 6c, the liquid cilia are arranged along a line to form a one-dimensional array along which the fluid can be pumped (Supplementary Movie 7). At $t = 0$ s, the tracer edible red dye solution is added from the left side while the ferrofluid droplets maintain the original elliptical

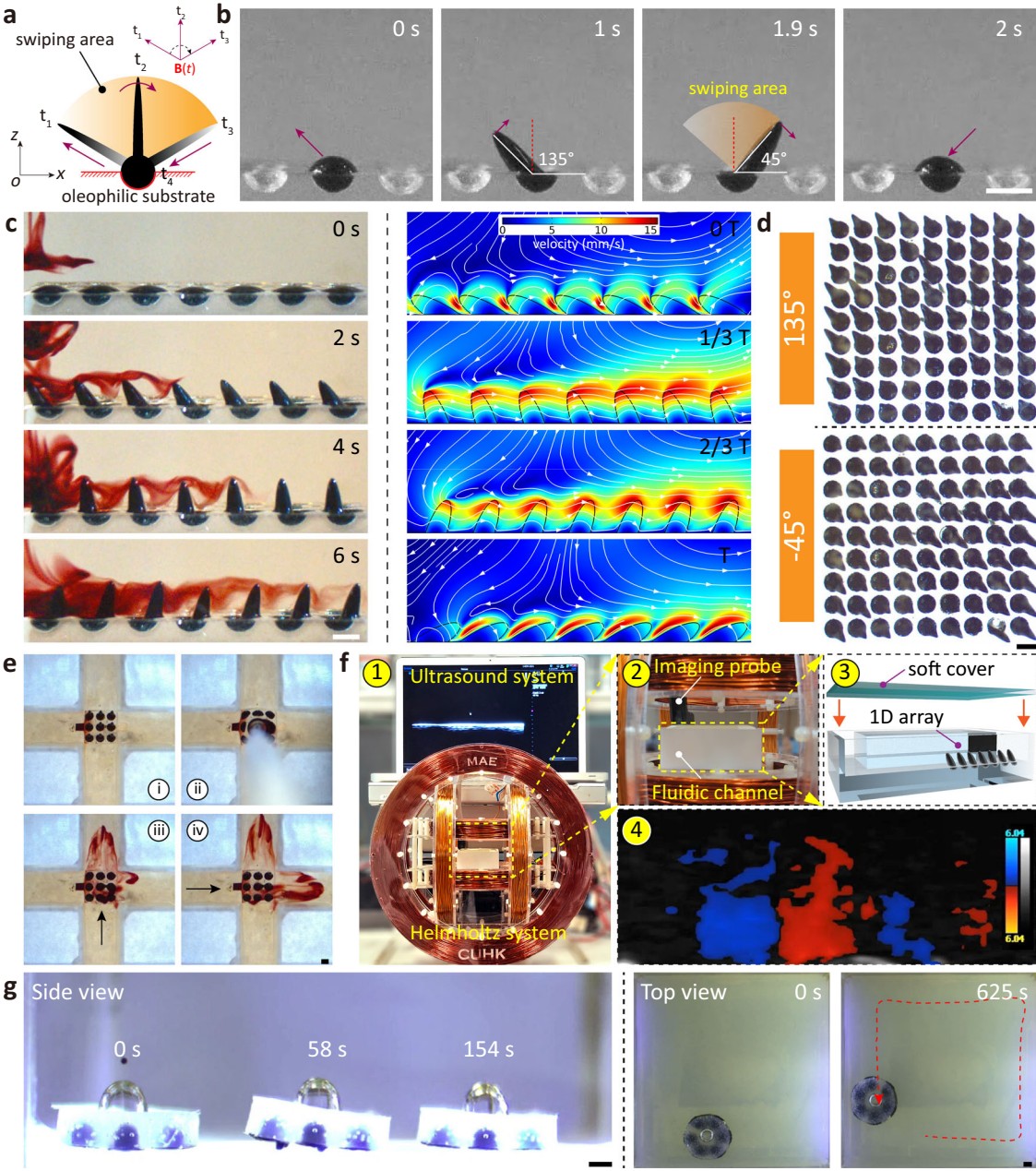

**Fig. 6 | Programmable liquid cilia as fluidic devices. a** Schematics of 2D non-reciprocal motion of single liquid cilium. **b** Snapshots of the nonreciprocal dynamics of single liquid cilium. **c** Experimental and simulated results of the fluid flow distribution of the liquid cilia array with synchronous waves. Color bar represents streaming velocities. **d** Top views of liquid cilia matrix with 2D synchronous waves. **e** Snapshots of a liquid cilia matrix pumping red food dye.

**f** Ultrasound doppler imaging visualization of liquid cilia array pumping porcine blood in enclosed channels. The experimental setup for blood transporting on liquid cilia arrays consists of a Helmholtz system with an ultrasound system. Color bar represents Doppler signal. **g** Demonstration of the potential of liquid cilia matrix as mobile soft robots. The substrate of the liquid cilia is a transparent resin material. Scale bars, 2 mm.

shape. When the magnetic field $\mathbf{B}_{Cilia}(t)$ is applied ($t = 2$ s), the ferrofluid droplets deform into liquid cilia exhibiting synchronous waves and pumps the edible red dye to the right. After 6 s, the edible dye is pumped from the left side to the right side with a fluid motion velocity of about 3.9 mm/s. Fluid field distribution during a whole oscillation cycle of the one-dimensional cilia array is simulated using COMSOL Multiphysics. The simulation results in Fig. 6c illustrate that the liquid cilia array generates directional flow along the oscillation direction during the 135°–45° oscillation, which can be used to pump liquids. The fluid velocity is related to the periodicity and beat frequency of the array is shown in Supplementary Fig. 26. The strength of the magnetic field determines the length of the liquid cilia, the size of the static liquid cilia can be elongated to 5 times the initial length (1 mm) under

the magnetic field of 18 mT (Supplementary Fig. 27a). In addition, the step-out frequency of liquid cilia based on ferrofluid droplets is determined by the magnetic strength. The cut-off frequency is 1.1 Hz when the magnetic field strength is 9 mT; when the magnetic field strength increases to 18 mT, the cut-off frequency is about 4.2 Hz (Supplementary Fig. 27b). And the pumping speed of liquid cilia is not determined by the frequency alone. We can also increase the pumping speed by adjusting the external magnetic field to increase the cilia length and oscillation angle. Supplementary Fig. 28 displays that a one-dimensional array of liquid cilia can rapidly pump liquid within a closed square channel. In addition, the direction of oscillation of liquid cilia in space can be adjusted arbitrarily compared to elastomer-based cilia, allowing the design and fabrication of 9 × 9 omnidirectional liquid cilia

matrix (Fig. 6d and Supplementary Movie 7). The oscillation direction of the omnidirectional liquid cilia matrix is mapped onto a plane and is tuned arbitrarily, such as oscillating along 135° or −45° and rotating 360° along the z-axis (Fig. 6d and Supplementary Fig. 29). This omnidirectional liquid cilia matrix can also transport solid cargo along a predesigned trajectory (Supplementary Fig. 30). In addition to transporting solid cargo, the omnidirectional liquid cilia matrix also controls the direction of the pumped fluid (Fig. 6e and Supplementary Movie 7). A 3 × 3 liquid cilia matrix exists in the middle of the cross-shaped flow channel, and then edible dyes are added in the middle, and the liquid matrix can pump fluid continuously in both directions. The combination of a three-axis Helmholtz coil system and an ultrasound imaging system (Terason t3200, Teratech Corporation, USA) allows wireless actuation and tracking of liquid cilia in confined narrow channels and their use for pumping fresh porcine whole blood (Fig. 6f). In addition, the liquid cilia are combined with a silicone elastomer sheet and placed upside down on a substrate, which uses traveling synchronized waves to generate motion (Supplementary Movie 8). As shown in the Fig. 6g, the bubbles can increase the buoyancy of the robot, and the robot moves at a speed of 0.3 mm/s under the segmented magnetic field $\mathbf{B}_{Cilia}(t)$ ($f = 2$ Hz, $B_m = 9$ mT, $\theta = 30°$). The robot can move in all directions (Fig. 6g: Top view). The advantage of using the ferrofluid as artificial cilia is the length of artificial cilia based on ferrofluid droplets can be adjusted compared to artificial cilia based on magnetic elastomers. The adjustable length of the liquid cilia allows the liquid cilia to be reconfigured in situ for more flexible fluid pumping. In addition, the performance of liquid cilia compared to other artificial cilia at low Reynolds (Re) numbers is shown in Table S1. The liquid cilia shown here provide novel motion mechanisms that can be used to create wireless microfluidic devices and as a robotic platform, that is of great value in fluidic applications.

## Miniature wireless liquid skin at total wetting

Soft robots prepared using soft materials such as silicone and polydimethylsiloxane combined with magnetic particles can achieve a variety of morphologies. However, once these soft robots are manufactured, their shape is usually fixed. Although these soft robots can perform specific tasks in unstructured environments, it is difficult to adapt to various structures and tasks. Therefore to increase the versatility of soft miniature robots, further simplified millirobot construction strategies are needed. Due to the solvent-philic nature of the silicone elastomer surface towards mineral oil, the oil-based ferrofluid can wet, absorb, and penetrate the silicone elastomer sheet, which can then be endowed with magnetic properties by immobilizing a layer of hard iron oxide-based nanoparticles on the surface. We demonstrate that ferrofluid droplets act as wireless liquid skin and that a series of miniature soft robots can be built through the adhesion strategy: walking spiderbot, crawling caterpillarbot, and swimming fishbot. As shown in Fig. 7a, the wireless liquid skin transforms a petal-shaped elastomeric sheet into a spider robot, maneuvering in a walking mode (Supplementary Movie 9). At $t = 0$ s, a 10 mm diameter ferrofluid droplet is 40 mm away from the elastomeric sheet. In order to make the ferrofluid droplets actively and uniformly adhere to the elastomeric sheet, a high-frequency magnetic field is applied to split ferrofluid droplet and then actuate sub-droplets to move to the target sheet ($t = 41$ s). After 145 s, the ferrofluid droplet is completely and uniformly coated onto the elastomeric sheet. The coating process is sequentially divided into three key stages. In the first stage, the ferrofluid droplet is driven by the magnetic field to reach the target elastomer and then, under the influence of the magnetic field, thoroughly wets the elastomer surface. In the second stage, ferrofluid penetrates the silicone elastomer surface layer. Due to the high mutual solubility of the hydrocarbon solvent and the elastomeric sheet, the solvent diffuses rapidly from the surface to the inner layer along its concentration gradient. As a result, the upper part of the elastomer sheet becomes swollen and inflated. At the same time, most dispersed nanoscale iron oxide particles concentrate and precipitate out, and some adhere to the surface. In the third stage, ferrofluid penetrates the body of the elastomer sheet. The solvent continues to evaporate, and the magnetic particles start to agglomerate. Iron oxide nanoparticles aggregate on the scanned surface and form a continuous rigid layer. The petal-shaped elastomeric sheet acquires magnetism and transforms into a spider robot that moves along a square trajectory driven by external magnetic field $\mathbf{B}_t(t)$ ($f = 4$ Hz, $B_m = 9$ mT). A side view of the spider robot locomotion is shown in Supplementary Fig. 31. In addition, the ferrofluid droplets also transform the elastomer sheet into a caterpillar robot and a fish robot. As shown in Fig. 7b, the ferrofluid droplet splits into two sub-droplets as needed and adheres to each end of the elastomer sheet (Supplementary Movie 9). Under the oscillating magnetic field, the robot crawls forward. A side view of its motion is shown in Supplementary Fig. 32. The ferrofluid droplet is attached to one end of the elastomer sheet to form a fish robot that swims under the oscillating magnetic field (Fig. 7c and Supplementary Movie 9). The propulsion mechanism for the fish robot is the same as the body-caudal fin propulsion swimming robot[52,53]. The head of the fish robot is firstly wetted by the ferrofluid droplet, which enables it to be controlled by the external magnetic field. The oscillating magnetic field will force the robot's head to sway from side to side, a process similar to excitation at one terminal of a beam at its first-order bending vibration frequency, thus causing the beam to bend. Since the head is heavier and more rigid (since the elastomer sheet at this region is wetted by the ferrofluid droplet), the tail displacement will be more pronounced. Then when the tail swings from side to side, the water on the rear side is continuously pushed away from the body, causing vortices behind its tail as the COMSOL Multiphysics simulation result in Supplementary Fig. 33, and the reaction force generated by the fluid on the robot's body during this process will eventually push the robot forward.

In addition, the ferrofluid as a movable skin can travel through a complex maze environment and reach the vicinity of the target silicone elastomer sheet. The ferrofluid absorbs into the silicone sheet and transforms it into a mobile robot that can move from the bottom to the water surface for controllable movements (Fig. 7d and Supplementary Movie 10). When traversing the complex maze environment, the wireless liquid skin can jump over fences in jumping mode and squeeze through narrow passages in stretching mode, respectively. After reaching the vicinity of the target silicone elastomer sheet, the large ferrofluidic droplets are controllably split into multiple sub-droplets. Then these sub-droplets evenly adhered to the surface of the silicone elastomer sheet, imbuing it with magnetic properties (Fig. 7d: the upper left). As shown in the Fig. 7d (the lower left), the magnetic droplet is split into a plurality of tiny droplets, which are moved to the elastic silicone sheet under the drive of the magnetic field, and the magnetic field drives the magnetic robot to perform controllable rolling. After being covered with ferrofluid, the magnetic silicone elastomer sheet still maintains hydrophobic properties, and then a vertical magnetic field is applied to make the elastomer stand. When a part of the magnetic elastomer sheet protrudes out of the water surface, the elastomer sheet will emerge from the water surface under the surface tension (Fig. 7d: the lower right and Supplementary Fig. 34). Ultimately, there is a controlled movement on the water (Fig. 7d: the upper right). In addition, when the robot is submerged in the IPA solution, the rigid layer of magnetic nanoparticles adhering to the robot surface is destroyed by stress concentration arising from a mismatch between the expansion ratio of the stiff layer and the silicone elastomer sheet. The magnetic nanoparticles are gradually disintegrated and enriched from the silicone elastomer sheet under the magnetic field (Supplementary Fig. 35). The magnetic particles can be reused to make ferromagnetic fluids, while the remaining silicone elastomer can be reused to build robots.

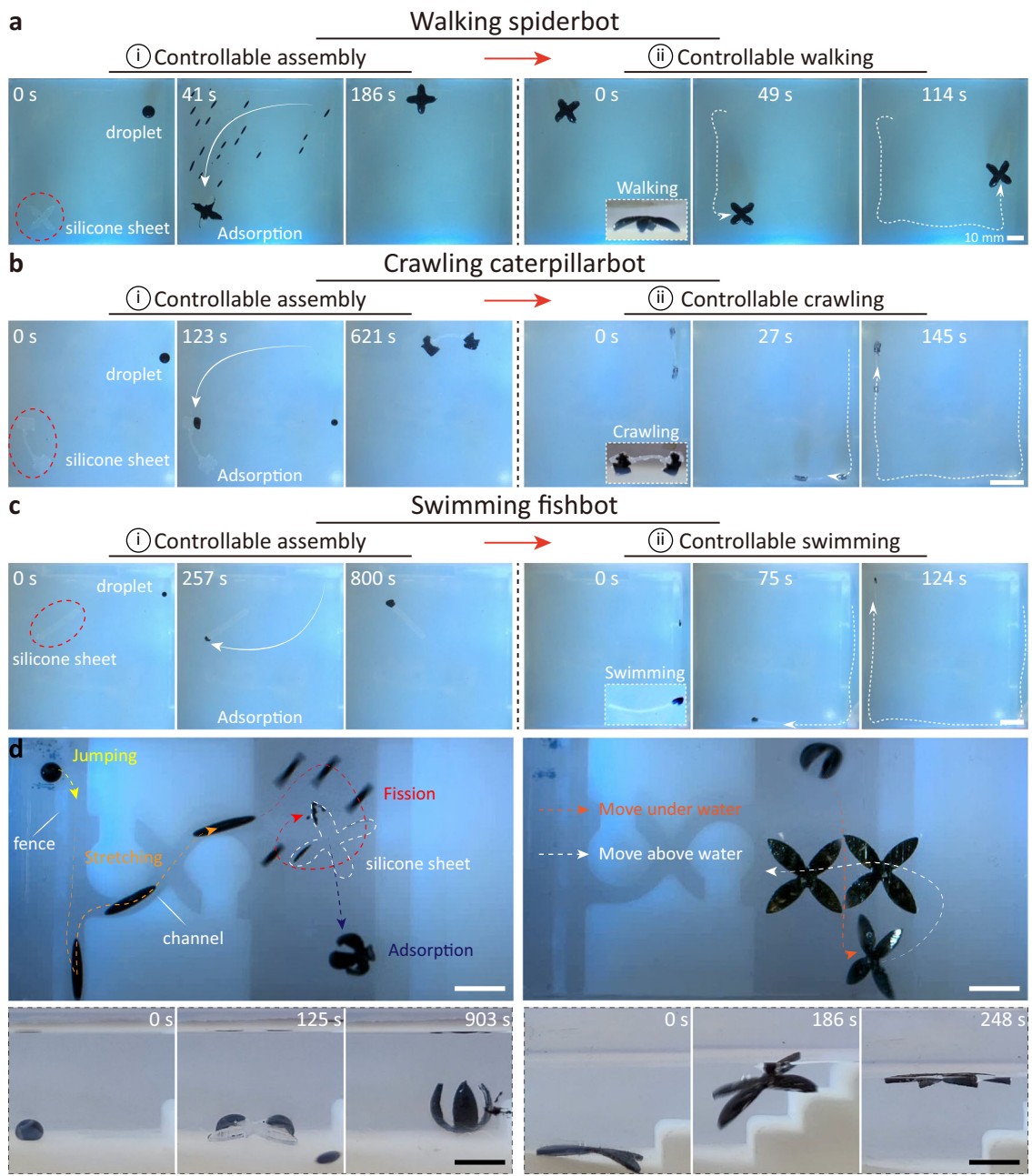

**Fig. 7 | Mobile liquid skin converts inanimate objects to miniature soft machines. a** The liquid skin turns silicone sheet into walking spiderbot. **b** The liquid skin turns silicone sheet into crawling caterpillarbot. **c** The liquid skin turns silicone sheet into swimming fishbot. **d** The upper left: the active liquid skin travels through a complex maze to reach the silicone sheet, which is transformed into a mobile robot through an adhesive strategy. The upper right: the robot moves from the bottom of the liquid to the surface of the liquid, and perform controlled movement on the surface. The lower left: the process of liquid skin adhering to the silicone sheet. The lower right: the process of the robot rising from the bottom to the surface. All the substrates are made of hydrogel. Scale bars, 10 mm.

In summary, our results show that ferrofluid droplets' reconfigurability and wetting properties can be integrated to construct multifunctional miniature soft machines. Ferrofluid droplet machines can be stimulated using programmed alternating magnetic fields in various motion modes: stretching, jumping, spinning, tumbling, kayaking, and oscillating. Compared to existing soft devices based on magnetically driven elastomers, our approach allows for more significant deformation (e.g., controlled splitting and fusion) in these droplet machines and enables complex shape deformation behavior on-demand. Jumping over high obstacles, crossing narrow channels, and maneuvering over changing textured surfaces demonstrate the high environmental adaptability of droplet machines. Compared with the ferrofluid droplets driven by the magnetic field gradient force[36], the magnetic

torque-driven ferrofluid droplets have various motion modes and split modes. When coupled with different surfaces to form assembled machines, they can also assume multiple roles under different task requirements, including acting as liquid capsules for solid or liquid cargo transport, two-dimensional cilia array matrix for pumping and agitating complex biological fluids, and intelligent skin for transforming inanimate objects into miniature soft machines. The external magnetic field can determine liquid cilia's length, step-out frequency, and rotation angle. If the Halbach array device with stronger magnetic field strength is used, the pumping speed of the liquid cilia can be further increased and drives the liquid cilia of the micrometer scale. In addition, the active wireless liquid skin can controllably navigate near inanimate targets and then transform into a soft machine through an

adhesive strategy. This dynamic and movable liquid skin has the advantage of selective adhesion.

In addition, the proposed approach to building multifunctional soft machines with the wetting properties of ferrofluids could inspire future machine construction. Other functional liquids, such as liquid metals, can also be introduced to construct miniature machines. On the one hand, similar to ferrofluids, liquid metal droplets can also act as liquid machines. On the other hand, solid-liquid coupled systems can also be constructed in combination with other solid materials, in which the wetting properties will play an important role. The solid-liquid coupled miniature machines have different behaviors under different wetting conditions. Moreover, the wetting properties of other solid interfaces can be programmed to change, and future work can focus on altering the wettability of substrates to achieve mobile liquid skin attachment and desorption. In general, the introduction of functional liquid materials in the construction of small machines adds functionality and adaptability, making them promising for future applications in fields such as biomedicine.

# Methods

## Materials

Hydrocarbon oil-based ferrofluids (purchased from Taobao, China) were used in the experiments, with a dynamic viscosity of 50 cP, a saturation magnetization of 43 mT, and a density of 1.29 g/ml. Acrylamide ($C_3H_5NO$, 99.0%), ammonium persulfate ($H_8N_2O_8S_2$), N,N'-Methylenebis(acrylamide) ($C_7H_{10}N_2O_2$, 99.0%), tetramethylethylenediamine (($CH_3$)$_2$NCH$_2$CH$_2$N-($CH_3$)$_2$), phosphate-buffered saline (PBS) were purchased from Aladdin Co. Ltd. (Shanghai, China) for constructing various terrains in experiments. Ecoflex 00-30 polymer matrix was purchased from Smooth-On Inc. Fresh bladder, gastrointestinal and biliary systems of pig were purchased from the local market. The placentas of the newborn babies used in the experiments were collected from the Prince of Wales Hospital, Hong Kong. The collection of human placentas was approved and overseen by The Joint Chinese University of Hong Kong-New Territories East Cluster Clinical Research Ethics Committee (The Joint CUHK-NTEC CREC) (Ref. No. 2020.384). All enrolled patients provided written informed consent. The human placenta was donated by pregnant women in collaboration with the Department of Obstetrics and Gynaecology (CUHK).

## Terrain construction

The continuous hurdles, stairs, holes in the walls, a circularly curved channel, a sharp turn, a gap, comb-like channels and maze terrain shown in the experiment were all manufactured using positive and negative casting. The various positive and negative molds are first printed using a 3D printer (RAISE3D Pro2) and a uniform layer is applied using Ecoflex 00-30 to make the mold surface more flat. The hydrogel solution was then configured, typically with 400 mg of acrylamide, 35 mg of N,N'-Methylenebis(acrylamide), and 15 mg of ammonium persulfate dissolved in 3 ml of DI water. The hydrogel solution was then added to a 50 mm × 50 mm × 50 mm experimental tank and tetramethylethylenediamine (1:100 volume ratio to the hydrogel solution) was added. Finally, the molds were added first in the tank and removed after the hydrogel had cured to obtain the desired topographical environment. The bile duct phantom used in the experiments was constructed from 3D printed soft gel material, with the real porcine biliary system fixed to the substrate to construct the narrow channel environment. In addition, fresh bladders and stomachs of pigs were dissected and fixed to the substrate to obtain an uneven inner surface of the bladder and an inner surface of the stomach filled with folds. Here, fresh bile ducts, bladders, and stomachs of pigs were purchased from the market, dissected and rinsed with water. To visualize the movement of ferrofluid droplets within the placental vasculature, the internal blood of the placenta was first drained and filled with PBS.

## Magnetic actuation system

There are two sets of magnetic actuation systems in the experiment. A Helmholtz electromagnetic coil setup consists of three sets of orthogonally placed coils, signals are generated by four motor drivers (ESCON70/10, Maxon) and controlled by a PC. By adjusting mathematical expressions into the control program, we can use this setup to generate 1D, 2D and 3D magnetic fields with specific requirements. When the system is matched with an ultrasonic device (Terason t3200, Teratech Corporation, USA), it can carry out real-time ultrasonic-guided ferrofluid droplet motion. In addition, a 6-DOF robotic arm and three coils are assembled together, and paired with a fluoroscopy imaging device (Artis zeego, SIEMENS) to guide the movement of ferrofluid droplets in real time. Magnetic field gradient force and magnetic torque can be generated in this system.

## Ferrofluid droplet shape in the uniformed magnetic field

The capillary and magnetic forces, the magnetostatic Maxwell equations and the incompressible Navier–Stokes equations form a dilute emulsion model for ferrofluid droplets that describe the dynamics and splitting behavior of ferrofluid droplets in two-phase fluids[42]:

$$\nabla \cdot (\mu_0 \zeta(\mathbf{x}) \nabla \psi) = 0 \tag{4}$$

where $\mu_0 \zeta$ is the magnetic permeability, $\psi$ is a scalar potential and $\mathbf{x}$ is the position vector. The Navier–Stokes equations as follows:

$$\nabla \cdot \mathbf{u} = 0 \tag{5}$$

$$\rho \frac{D\mathbf{u}}{Dt} = -\nabla p + \nabla \cdot \left[ \lambda(\mathbf{x}) \eta (\nabla \mathbf{u} + \nabla \mathbf{u}^T) \right] + \mathbf{F}_m + \mathbf{F}_s \tag{6}$$

where $\rho$ is the density, $\mathbf{u}$ is the velocity vector, $p$ is the pressure, $\mathbf{F}_m$ is the magnetic force, and $\mathbf{F}_s$ is the capillary force. The magnetic force is defined as

$$\mathbf{F}_m = \mu_0 \mathbf{M} \cdot \nabla \mathbf{H} = \mu_0 (\zeta(\mathbf{x}) - 1) \mathbf{H} \cdot \nabla \mathbf{H} \tag{7}$$

and the capillary force is given by

$$\mathbf{F}_s = -\sigma K \delta_s \mathbf{n} \tag{8}$$

where $\mathbf{H}$ is the magnetic field, $K$ is the curvature, $\sigma$ is the interfacial tension, $\delta_s$ is a Dirac delta distribution, and $\mathbf{n}$ is the unit normal vector.

The stretching length of the ferrofluid droplet under the uniformed magnetic field is determined by the surface stress tensor and the interaction between the magnetic field and dipole moments of nanoparticles. For the inviscid, isothermal, and incompressible ferrofluid droplet, the steady-state flow can be expressed by using the ferrohydrodynamic (FHD) Bernoulli equation[47,48]:

$$p^* + \frac{1}{2}\rho v^2 + \rho g h - p_m = \text{const} \tag{9}$$

with the boundary conditions:

$$p^* + p_n = p_0 + p_c \tag{10}$$

where $\rho$ is the density, $v$ is velocity, $g$ is the acceleration of gravity, $h$ is the elevation from a reference level. $p_m = \mu_0 \int_0^H M dH$ is the fluid-magnetic pressure with $\mu_0$, $H$ and $M$ indicating the vacuum permeability, the magnetic field intensity, and the fluid magnetization, respectively. $p^* = p + \mu_0 \int_0^H [\partial(Mv)/\partial v]_{H,T} dH$ is the composite pressure in the ferrofluid with $p$, $v$, and $T$ indicating the thermodynamic pressure, the specific volume ($v = \rho^{-1}$), and temperature, respectively. $p_n = \frac{\mu_0}{2} M_n^2$ is the magnetic normal traction with $M_n$ denoting the fluid

magnetization component normal to the fluid surface. $p_0$ pressure in the nonmagnetic fluid. $p_c = 2C\sigma$ is the capillary pressure with $C$ and $\sigma$ indicating the radius of curvature and the surface tension. For example, when there is no magnetic field, the FHD Bernoulli equation turns to $p_1^* + \rho g h_1 = $ const. When a vertical uniform magnetic field is applied, the FHD Bernoulli equation becomes $p_2^* + \rho g h_2 - \mu_0 \int_0^H M dH = $ const. According to the boundary condition, $p_1^* = p_0$ and $p_2^* = p_0 - \frac{\mu_0}{2} M^2$ ($M_n = M$ in this case). Then we have

$$\Delta h = h_2 - h_1 = \frac{1}{\rho g} \left( \mu_0 \int_0^H M dH + \frac{\mu_0}{2} M^2 \right) \tag{11}$$

$\Delta h$ with a vertical uniform magnetic field applied, indicating that the ferrofluid droplet rises vertically. Similarly, when the uniform magnetic field points in other directions, the ferrofluid droplet would also elongate along the direction of the magnetic field.

## Fabrication of liquid cilia
Using 3D printing technology, arrays of $1 \times 7$, $3 \times 3$ and $9 \times 9$ holes were made from transparent resin material with a hole size of 2 mm, depth of 1 mm and a center spacing of 4 mm, which was then submerged under water and subsequently injected with a ferromagnetic fluid, $3 \mu L$ per hole. The droplets of ferrofluid are held on top of the substrate by their adhesion to the resin material and, under the external magnetic field, the droplets are transformed into liquid cilia.

## Reporting summary
Further information on research design is available in the Nature Portfolio Reporting Summary linked to this article.

# Data availability
All data needed to evaluate the conclusions in the paper are present in the paper and the supplementary materials. Additional data related to this paper may be requested from the authors. Source data are provided with this paper.

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

## Acknowledgements

The research work is financially supported by the Hong Kong Research Grants Council (RGC) with project Nos. RFS2122-4S03, R4015-21, C1134-20GF and E-CUHK401/20; the ITF project with Project No MRP/036/18X funded by the HKSAR Innovation and Technology Commission (ITC); the Croucher Foundation Grant with Ref. No. CAS20403, and the CUHK internal grants. The authors also thank the support from Multi-Scale Medical Robotics Centre (MRC), InnoHK, at the Hong Kong Science Park, and the SIAT-CUHK Joint Laboratory of Robotics and Intelligent Systems.

## Author contributions

M.S. conceived and designed the research. M.S. performed all the experiments and wrote the manuscript. M.S., B.H., and S.Y. performed the simulations and analyzed the data. M.S., B.H., and X.W. fabricated the structures. L.Z. and C.M. mentored the work and revised the manuscript. All authors contributed to the editing of the manuscript. We thank Dr. K.F.C. for preparing the placenta of the newborn baby as an unstructured lumen environment. M.S. and B.H. contributed equally to this work.

## Competing interests

The authors declare no competing interests.
