## [Peer Review File · Nature Communications]

REVIEWER COMMENTS

Reviewer #1 (Remarks to the Author):

Sun et al. investigated how ferrofluids could wet different substrate surfaces including hydrogel, PDMS, elastomer, metal etc. to realize several functions including artificial cilia, capsule for carrying liquid cargos, and ferrofluid skin for elastomer robots. Although the demonstrations look interesting, they are only replicating the many demonstrations from recently published works. The overall feeling is that this work does not provide a fundamental new mechanism except some discretized example showcases without justification of the advantages over previous devices. Therefore, it is hard to convince the reviewer that the novelty is sufficient to advance the field. Some further comments are listed as below.

1. The novelty of this work is very limited. The wetting properties of ferrofluid droplets using hydrocarbon oil seem to be straight forward and have already been understood in multiple works [1,3,4].

2. In addition, the demonstrations of this work have been shown in many recently published works in the fields. For example, the cilia demonstration is based on the works from [2] and many other works on magnetic cilia. The splitting and merging, environmental adaptive behaviors, and group motions of ferrofluid droplets have been shown in [3,4]. The coating of the ferrofluid on elastomers is just like those shown in [5]. This work lacks the depth and uniqueness considering all these mentioned publications in the field.

3. The authors mentioned that “ferrofluid droplets’ reconfigurability and wetting properties can be programmed” so that different functions could be realized in the conclusion. However, the wetting ability of ferrofluid on different surfaces depends on the substrate surfaces, which makes it very misleading to say “programmed”. This also raises the question why people should use these ferrofluids as liquid capsule, artificial cilia, and coating elastomer structures, considering other devices/robots could already demonstrate similar or even better performances.

4. Did the authors measure the contact angles of hydrocarbon oil on different surfaces? How are they different from the ferrofluid droplets using hydrocarbon oil as a carrier fluid?

5. In Fig. 2B, what is the magnetic field strength? The figure only shows the normalized magnetic field. The absolute value needs to be provided.

6. The surface conditions in each demonstration and any surface treatment needs to be clearly stated in the figure captions and associated texts.

References.

[1] Wang, W., Timonen, J.V., Carlson, A., Drotlef, D.M., Zhang, C.T., Kolle, S., Grinthal, A., Wong, T.S., Hatton, B., Kang, S.H. and Kennedy, S., 2018. Multifunctional ferrofluid-infused surfaces with reconfigurable multiscale topography. *Nature*, 559(7712), pp.77-82.

<https://www.nature.com/articles/s41586-018-0250-8>

[2] Gu, H., Boehler, Q., Cui, H., Secchi, E., Savorana, G., De Marco, C., Gervasoni, S., Peyron, Q., Huang, T.Y., Pane, S. and Hirt, A.M., 2020. Magnetic cilia carpets with programmable metachronal waves. *Nature communications*, 11(1), pp.1-10.

<https://www.nature.com/articles/s41467-020-16458-4>

[3] Fan, X., Dong, X., Karacakol, A.C., Xie, H. and Sitti, M., 2020. Reconfigurable multifunctional ferrofluid droplet robots. *Proceedings of the National Academy of Sciences*, 117(45), pp.27916-27926.

<https://www.pnas.org/doi/abs/10.1073/pnas.2016388117>

[4] Fan, X., Sun, M., Sun, L. and Xie, H., 2020. Ferrofluid droplets as liquid microrobots with multiple deformabilities. *Advanced Functional Materials*, 30(24), p.2000138.

<https://onlinelibrary.wiley.com/doi/full/10.1002/adfm.202000138>

[5] Yang, X., Shang, W., Lu, H., Liu, Y., Yang, L., Tan, R., Wu, X. and Shen, Y., 2020. An agglutinate magnetic spray transforms inanimate objects into millirobots for biomedical applications. *Science robotics*, 5(48), p.eabc8191.

<https://www.science.org/doi/10.1126/scirobotics.abc8191>

Reviewer #2 (Remarks to the Author):

The manuscript reports the results for the design of soft millimeter-sized robots by harnessing the wettability and reaction to external magnetic fields of ferrofluid droplets for controlled reconfigurability. The ferrofluid droplet at low wetting exhibits multimodal motions, such as stretching, jumping, spinning, tumbling, kayaking, and wobbling. Also, a single ferrofluid droplet can be controlled to split into multiple sub-droplets and then re-fuse back into a single droplet. The soft droplet can traverse various terrains and textures in unstructured environments. In addition, the ferrofluid droplets can be configured as a liquid capsule (at low wetting), enabling active liquid and solid cargo delivery, as a liquid cilia matrix (at high wetting), capable of pumping fluids, and as a liquid skin (at complete wetting), thereby facilitating the construction of multiple types of soft machine by adhesion to elastic materials.

The idea of using ferrofluids for the construction of miniature soft machines is very interesting, but not entirely new (see, e.g. Refs. [36-41]). As far as I can see, the main new aspect of the current manuscript is to also consider the wetting of various substrates by a ferrofluid droplet. This is an important aspect, which strongly increases the number of possibilities for the construction of miniature soft machines. Thus, this work improves the achievable complexity of small magnetic soft machines and boosts their future capabilities for robotics and biomedical engineering applications.

This is a "can do" manuscript, in which many kinds of manipulation of ferrofluids

by time- and space-dependent magnetic field are demonstrated. This comes at the cost of often not very detailed explanations of the underlying mechanisms.

The authors should address the following questions and comments:

(1) p.3: As explained above, the main new aspect of the current study is wettability. However, several part of the manuscript also concern aspects which are not related to wetting. It would thus be good to explain in a bit more detail how the current study is distinguished from Refs. [36-41].

(2) p.3: When the authors mention "low wettability", it is not obvious of what: the cargo? a substrate?

(3) p.3: The authors state that "In-depth studies have NOT yet been carried out to characterize the wetting dynamics of ferrofluid droplets ...". I find this a bit surprising, after many years of intensive studies of ferrofluids. In fact, the authors themselves cite Refs. [44-48] -- for not obvious reasons these are only mentioned in the Methods section! In addition there are, e.g. the publications -- S.Shyam et al., Coll. Surf. A 586, 124116 (2020); -- V.A. Roodan et al., Soft Matter, 16, 9506-9518 (2020).

(4) p.3: The authors claim that "The ferrofluid droplets are highly adaptable to both structured and unstructured environments ...". These are many claims! Are all of them substantiated in the manuscript?

(5) p.4: What are "wireless" liquid capsules?

(6) p.6: Wetting properties depend on three surface tensions! What is the "suspending" phase? I guess water. Furthermore, it seems to me that as the carrier fluid of a ferrofluid is "oily", the wetting properties should mainly determined by the hydrophobicity of the substrate.

(7) p. 11: No wetting seems to be involved in Sec. 2.3.

(8) p. 11: Medical robots: Are ferrofluids sufficiently biocompatible?

Where do they operate? Clearly the droplets would be too large for blood vessels.

(9) p.12: The authors state that "... inside an annular channel, the ferrofluid droplet constantly stretches and shrinks to achieve locomotion by utilizing the frictional force of the sidewall." I don't understand this argument. What breaks the symmetry between clockwise and counter-clockwise motion?

(10) p.14: The switching between different modes of operation in a heterogeneous environment requires detailed knowledge about instantaneous location and environment! Can this be achieved in medical applications?

(11) p.14: Syringe injection of a cargo liquid: Seems to require complete wetting by the ferrofluid to avoid instability and ejection!

(12) p.19: Are the anchoring sites of the cilia little dimples in the substrate? Otherwise, why would the ferrofluid droplets not move or be displaced on the substrate?

(13) p.18: I don't agree with the statement "Unlike natural or other artificial cilia, this presented liquid cilia based on ferrofluid exhibit nonreciprocal motion, even though they do not have a power stroke or a recovery stroke." Of course, both types of strokes are present.

(14) p.20: How effective are the cilia arrays in pumping fluid? How does the fluid velocity of 3.9 mm/s relate to the periodicity of the array and the beat frequency? How does this compare to other artificial cilia?

(15) p.21: The ferrofluid penetrates the body of the elastomer sheet -- I would not really denote this as wetting!

(16) p.22: How is the spider structure (with four legs) obtained from a uniform coverage?

(17) p.22: "... fish robot that swims under the oscillating magnetic field"? The propulsion mechanism is not explained! Is this a scallop-type propulsion at finite Reynolds number?

Reviewer #3 (Remarks to the Author):

This manuscript demonstrates the versatility of ferrofluid droplets for robotic applications. The authors present interesting applications that could be of relevance to soft robots. I believe this manuscript would be of interest to the readership of the Nature Communications and would be happy to recommend it for publications; I only have a few minor comments/questions:

1- For the biomedical applications, can the authors comment on a) biocompatibility of the ferrofluids, and whether it would be ok for the fluids to spread through the tissues, etc, and b) whether the use of a magnet given the required distance and imaging, etc is a realistic way of guiding ferrofluids in medical applications?

2- in Fig. 3, why do the drops coalesce?

3- can you elaborate on the mechanism of drug release in Fig. 5? If the contraction of the liquid droplet is causing the release, how come this release is not expected during the locomotion stages before the droplet gets to the target?

4- Related to liquid cilia in Fig. 6, qualitative arguments related to Scallop theorem are mentioned; a rough estimate of Reynolds number, however, shows that the flow is likely at $Re > 1$, and these arguments may not be relevant.

Response to Reviewer #1

Sun et al. investigated how ferrofluids could wet different substrate surfaces including hydrogel, PDMS, elastomer, metal etc. to realize several functions including artificial cilia, capsule for carrying liquid cargos, and ferrofluid skin for elastomer robots. Although the demonstrations look interesting, they are only replicating the many demonstrations from recently published works. The overall feeling is that this work does not provide a fundamental new mechanism except some discretized example showcases without justification of the advantages over previous devices. Therefore, it is hard to convince the reviewer that the novelty is sufficient to advance the field. Some further comments are listed as below.

Response: We thank the reviewer for their careful review and positive feedback on our work, such as "the demonstrations look interesting". Based on the reviewer's comments, we have substantially revised our manuscript by clarifying the contribution, deleting the overstated claims, adding more experimental data to support the statement, and reorganizing/rewriting the paper. Please check the following point-by-point response for details.

1. The wetting properties of ferrofluid droplets using hydrocarbon oil seem to be straight forward and have already been understood in multiple works [1,3,4].

Response: We are grateful for the reviewer's comments. These mentioned works by the reviewer indeed advanced the field of ferrofluid research and inspired us^[1-3], and they have been added and acknowledged in the revised manuscript (**please see references 36, 39, and 51**) because of the authors' significant contributions in this field. Wang et al. made a hierarchical magneto-

responsive composite surface by fully infiltrating the ferrofluid into a microstructured matrix and demonstrated several interesting features at three different length scales driven by magnetic field gradient forces: self-assembly of colloidal particles at the micrometer scale; regulated flow of liquid droplets at the millimeter scale; and switchable adhesion and friction, liquid pumping and removal of biofilms at the centimeter scale^[1]. Fan et al. studied the centimeter-scale ferrofluid droplets' motion behavior and the micron-scale ferrofluid droplets' assembly behavior, respectively^[2,3]. However, we can find significant differences between the previously reported literature and our work. Firstly, Wang et al. studied the performance of ferrofluids as functional surfaces at completely wetting under magnetic field gradient forces. The dynamics and potential application scenarios of ferrofluid droplets under magnetic torque still need further investigation. Secondly, although Fan et al. studied the dynamic behavior of centimeter-scale and micro-scale ferrofluid droplets, they were all under non-wetting conditions. The related work mentioned by the reviewer focuses on the fluidity and paramagnetic property of ferrofluids, and the related work mentioned by the reviewer focuses on the fluidity and paramagnetic property of ferrofluids under magnetic field gradient force, and the dynamic behavior of ferrofluid droplets under different wetting conditions under magnetic torque has not been fully investigated.

Here, we are more focused on studying the dynamic wetting behavior of ferrofluid droplets under magnetic torque, exploiting their wetting properties to construct complex miniature soft machines, and building bridges between fluidic materials and soft robotics. Such work builds on a broader range of efforts that draw inspiration from creatures in nature to develop a range of representative soft robots^[4]. To approach the advanced characteristics of natural organisms, these robotic systems are typically composed of materials with low Young's modulus or high deformability, such as polydimethylsiloxane^[5], shape

memory alloy^[6], and dielectric elastomer^[7]. However, continued progress in developing soft robots requires further exploration of fluidic materials for actuation, mechanical adaptability, and intelligent control. Fluids are an essential component of biological creatures, permeating all aspects of our lives, and also play a crucial role in the energy transfer^[8], motion regulation^[9], etc., of soft robotic systems. Here we show how harnessing the wettability of ferrofluid droplets allows for controlled reconfigurability and the ability to create highly versatile fluidic soft machines. Our research provides a paradigm for integrating liquids into tiny soft-bodied machines.

In short, inspired by these previous works, we have developed a method to construct soft-bodied machines via the wettability of ferrofluid, which bridges fluidic materials and soft robotics. Moreover, we have added new results in the revised manuscript and supporting information to highlight the innovation.

2. In addition, the demonstrations of this work have been shown in many recently published works in the fields. For example, the cilia demonstration is based on the works from [2] and many other works on magnetic cilia. The splitting and merging, environmental adaptive behaviors, and group motions of ferrofluid droplets have been shown in [3,4]. The coating of the ferrofluid on elastomers is just like those shown in [5]. This work lacks the depth and uniqueness considering all these mentioned publications in the field.

Response: We thank the reviewer for the comments and understand the concerns. We try to elucidate the differences and uniqueness of our work in the dedicated sections below. As the reviewer noted, to tackle the challenges of miniature soft robots in multifunction, materials design, motion control, and assembly tasks, many research efforts have developed a variety of soft robots,

such as magnetic cilia based on the elastomer^[14,15], transformable robots^[2,3,5], and the agglutinate magnetic spray^[16]. Our work is also an effort to address these challenges and has the uniqueness that distinguishes it from the previously reported literature:

(i) Regarding the cilia demonstration, our pumping mechanism of liquid cilia is entirely different from solid cilia^[10-15]. Natural cilia or artificial cilia based on magnetic elastomers usually perform buckling-based motion with two strokes, i.e., the power and recovery strokes result in a nonzero swiping area to achieve the liquid pumping^[11-15]. The ferrofluid droplet first uses its wetting property to the resin substrate and then uses its deformability, elongation-rotation-contraction, to achieve liquid pumping. Furthermore, in the absence of an applied magnetic field, the ferrofluid droplets immobilized on the substrate always maintain a spherical state, unlike solid cilia that maintain a rigid shape^[10-15], thus exerting less influence on the fluid flow. The performance of liquid cilia compared to other artificial cilia at low Re is shown in **Table R1.1**, which has been added in the revised manuscript as Table S1 (**please see section 2.5 and page 19**).

References	Normalized velocity	Fluid velocity	Driving frequency	Cilia length	Cilia number	Materials
	v/fL	v	f	L	N	
10	0.1	3.3 $\mu\text{m/s}$	1 Hz	31 μm	3×3	Superparamagnetic microparticles
11	0.143	500 $\mu\text{m/s}$	7 Hz	500 μm	1×6	Polydimethylsiloxane
12	0.0214	75 $\mu\text{m/s}$	10 Hz	350 μm	3×3	Polydimethylsiloxane

13	0.011	9 $\mu\text{m/s}$	34 Hz	25 μm	3000	Silicone elastomer
14	0.25	83 $\mu\text{m/s}$	0.083 Hz	4 mm	8 \times 8	Silicone elastomer
15	0.38	0.95 mm/s	2.5 Hz	1 mm	6 \times 6	Silicone elastomer
This work	0.147	0.195 mm/s	1.1 Hz	1.2 mm	1\times7	Ferrofluid

Table R1.1. Comparison of the fluid pumping performance using artificial cilia with literature. The normalized velocity, shows how fast the flow it can generate by one beating cycle.

(ii) Our work on the splitting and merging behavior, adaptive environmental behavior, and collective motion behavior of ferrofluid droplets also differ. The previously reported splitting and merging of ferrofluid droplets are generated under the action of magnetic field gradient force, and the resulting splitting behavior pattern is limited^[2]. Our work focuses on the split-fusion behavior of ferrofluids under magnetic torque. Since the magnetic torque can be programmed, we can achieve controllable splitting of the ferrofluid, such as splitting along a line or over to the plane. Furthermore, due to the limitation of magnetic field gradient forces, ferrofluid droplets have only a single motion mode, such as stretching, and thus their overall environmental adaptability is limited. Our work's magnetic field torque-driven ferrofluid droplets have multiple motion modes, such as stretching, jumping, spinning, tumbling, kayaking, and wobbling. The ferrofluid droplet can not only traverse narrow channels as previously reported, but also achieve crossing obstacles in the jumping mode. The micro-scale ferrofluid droplets used in the study of collective motion are prepared by ultrasonic dispersion, which cannot produce large deformations in the two-phase fluid and cannot be merged under the

magnetic torque, so permanent magnets must be used^[3]. The entire split and fusion process is discrete and cannot be precisely controlled.

(iii) Regarding the magnetic coating demonstration, unlike the magnetic spray created by Yang et al.^[16], our ferrofluid droplet has multimodal motion capabilities that act as active, movable “skin”. For example, this liquid skin can enter a narrow environment and then use its adhesive properties to enable inanimate objects to move by coating them with a thin, magnetically drivable film. This mobile liquid skin can not only endow inanimate objects with the ability to move but also can be flexibly manipulated remotely to take out target objects.

Overall, we construct soft-bodied machines through the wettability of ferrofluids and integrate magnetic cilia, multimodal motion, controllable splitting and fusion, environmental adaptability, and liquid skin into ferrofluidic systems. Previous work has mainly focused on the flow properties of ferrofluids, and the application scenarios of wetting properties with different interfaces have not been adequately explored, and our work makes up for this deficiency.

3. The authors mentioned that “ferrofluid droplets” reconfigurability and wetting properties can be programmed” so that different functions could be realized in the conclusion. However, the wetting ability of ferrofluid on different surfaces depends on the substrate surfaces, which makes it very misleading to say “programmed”. This also raise the question why people should use these ferrofluids as liquid capsule, artificial cilia, and coating elastomer structures, considering other devices/robots could already demonstrate similar or even better performances.

Response: We thank the reviewer for the comments and understand the concerns. We agree with the reviewer that the wetting ability of ferrofluids on different surfaces depends on the substrate surface. Our work did not investigate the programming of the wettability of substrate surfaces, so we corrected the original statement to avoid being misleading. The description "Our results show that ferrofluid droplets' reconfigurability and wetting properties can be programmed to construct multifunctional miniature soft machines to address environmental changes and multitasking requirements" has been modified as follows: "Our results show that ferrofluid droplets' reconfigurability and wetting properties can be integrated to construct multifunctional miniature soft machines to address environmental changes and multitasking requirements" **(please see section 3 and page 25)**. We will focus on developing substrates with switchable adhesion to oil-based ferrofluids^[17-19], and achieving the controllable attachment and desorption of mobile liquid skin in our future work. We thank the reviewer for giving us an excellent idea for our future work.

As the reviewer noted, other robots did demonstrate similar performance, but there are still some major differences between the previously reported literature and our work. According to your suggestion, we have added the comparison in the "Introduction" part of the revised manuscript **(please see section 1 and pages 3, 4 and 5)**. We list the main differences as follows for your reference:

(i) Regarding the magnetic capsule, the liquid capsules driven by magnetic field gradient forces generated by an array of planar magnetic coils cannot adapt to complex environments and have a single mode of motion^[2]. The solid capsules are prepared from silicone elastomers^[20]; the fabrication method is too complicated and has a limited amount of deformation to traverse cavities when the capsule size is comparable with the cross-sectional dimension of these confined spaces.

(ii) Regarding the artificial cilia, conventional artificial cilia are in solid state and their morphology cannot be easily changed^[10-15], so they maintain a rigid structure in the absence of a magnetic field, which may affect fluid flow (**Figure R1.1**). One advantage of our liquid cilia is that, in the absence of an applied magnetic field, they will shrink to the bottom of the substrate and take on a spherical shape, thus reducing the impact on fluid flow. Furthermore, this mechanism of pumping through elongation-rotation-contraction is beneficial to inspire the creation of new types of cilia.

Figure R1.1. Comparison of solid artificial cilia and liquid cilia in the presence and absence of external field.

(iii) Concerning the magnetic coating, unlike the passive magnetic spray created by Yang et al.^[16], our ferrofluid droplet has multimodal motion capabilities that act as active, movable skin. For example, this liquid skin can enter a confined

space and then use its adhesive properties to enable inanimate objects to move by coating them with a thin, magnetically drivable film. This mobile liquid skin can not only endow inanimate objects with the ability to move but also can be flexibly manipulated remotely to take out target objects.

In short, inspired by these previous works, we have developed a method to construct soft-bodied machines via the wettability of ferrofluid, which enables a variety of representative functions such as liquid capsules, artificial cilia, and coating elastomer structures in one ferrofluidic system. We are more focused on exploring the application scenarios of different interfacial wetting properties of ferrofluid droplets. Our research could inspire the integration of fluidic materials into miniature soft machines.

4. Did the authors measure the contact angles of hydrocarbon oil on different surfaces? How are they different from the ferrofluid droplets using hydrocarbon oil as a carrier fluid?

Response: We thank the reviewer for the comments. According to your suggestion, we have added the experimental results of contact angles of hydrocarbon oil on different surfaces in the revised manuscript (**please see page 7 and Figure S2**), which is also provided here for your reference. Since hydrocarbon oil is less dense than water, it is hard to measure the contact angle between the hydrocarbon oil and various interfaces in the underwater environment. So here, we measure and compare the contact angles of hydrocarbon oil and ferrofluid with multiple interfaces in the air environment. As shown in **Figure R1.2**, whether it is hydrocarbon oil or ferrofluid, the contact angle of the interface with hydrogel, glass, resin, PMMA, and metal in the air environment is less than 15°. In contrast, their contact angle with silicone is larger than 30°. However, the experimental results show that the contact angle

of ferrofluid with glass and hydrogel substrate is smaller than that of hydrocarbon oil with the interface of glass and hydrogel. In comparison, the contact angle of ferrofluid with silicone is larger than that of hydrocarbon oil with silicone substrate. This difference in contact angle may be due to different constituents. Unlike hydrocarbon oil, ferrofluids also contain magnetic nanoparticles and surfactants. In addition, the contact angle of hydrocarbon oil or ferrofluid with various interfaces in the air is different from that of ferrofluid in an underwater environment. This is because the hydrophobicity of the substrate mainly determines the underwater wetting properties.

Figure R1.2. The wetting properties between hydrocarbon oil, ferrofluid and different surfaces in the air environment. Scale bars, 1 mm.

5. In Fig. 2B, what is the magnetic field strength? The figure only shows the normalized magnetic field. The absolute value needs to be provided.

Response: We are grateful for the reviewer's expert advice and kind reminding. In the revised manuscript, the absolute value in Fig. 2B has been provided (the maximum value: 9 mT and the minimum value: magnetic field off). **Please see page 6 and Figure 2B of the revised manuscript for more details.**

6. The surface conditions in each demonstration and any surface treatment needs to be clearly stated in the figure captions and associated texts.

Response: We appreciate the reviewer's valuable suggestions. According to your suggestion, we have clearly stated the surface conditions and surface treatment in the figure captions and associated texts. **Please see pages 6, 8, 12, 17, 20, and 24 of the revised manuscript for more details.**

References

1. Wang, W., Timonen, J. V., Carlson, A., Drotlef, D. M., Zhang, C. T., Kolle, S., Grinthal, A., Wong, T., Hatton, B., Kang, S., Kennedy, S., Chi, J., Blough, R., Sitti, M., Mahadevan, L., Aizenberg, J. Multifunctional ferrofluid-infused surfaces with reconfigurable multiscale topography. *Nature* **2018**, 559, 77-82.

2. Fan, X., Dong, X., Karacakol, A., Xie, H., Sitti, M. Reconfigurable multifunctional ferrofluid droplet robots. *Proc. Natl. Acad. Sci. U. S. A.* **2020**, *117*, 27916-27926.
3. Fan, X., Sun, M., Sun, L., Xie, H. Ferrofluid droplets as liquid microrobots with multiple deformabilities. *Adv. Funct. Mater.* **2020**, *30*, 2000138.
4. Rus, D., Tolley, M. T. Design, fabrication and control of soft robots. *Nature* **2015**, *521*, 467-475.
5. Hu, W., Lum, G. Z., Mastrangeli, M., Sitti, M. Small-scale soft-bodied robot with multimodal locomotion. *Nature* **2018**, *554*, 81-85.
6. Laschi, C., Cianchetti, M., Mazzolai, B., Margheri, L., Follador, M., Dario, P. Soft robot arm inspired by the octopus. *Adv. Robot.* **2012**, *26*, 709-727.
7. Chen, Y., Zhao, H., Mao, J., Chirarattananon, P., Helbling, E. F., Hyun, N. S. P., Clarke, D., Wood, R. J. Controlled flight of a microrobot powered by soft artificial muscles. *Nature* **2019**, *575*, 324-329.
8. Aubin, C. A., Choudhury, S., Jerch, R., Archer, L. A., Pikul, J. H., Shepherd, R. F. Electrolytic vascular systems for energy-dense robots. *Nature* **2019**, *571*, 51-57.
9. Wehner, M., Truby, R. L., Fitzgerald, D. J., Mosadegh, B., Whitesides, G. M., Lewis, J. A., Wood, R. J. An integrated design and fabrication strategy for entirely soft, autonomous robots. *Nature* **2016**, *536*, 451-455.
10. Vilfan, M., Potočnik, A., Kavčič, B., Osterman, N., Poberaj, I., Vilfan, A., Babič, D. Self-assembled artificial cilia. *Proc. Natl. Acad. Sci. U. S. A.* **2010**, *107*, 1844-1847.

11. Rockenbach, A., Schnakenberg, U. The influence of flap inclination angle on fluid transport at ciliated walls. *J. Micromechanics Microengineering* **2017**, *27*, 015007.
12. Zhang, S., Wang, Y., Lavrijsen, R., Onck, P. R., den Toonder, J. M. Versatile microfluidic flow generated by moulded magnetic artificial cilia. *Sensors Actuators B Chem.* **2018**, *263*, 614-624.
13. Shields, A. R., Fiser, B. L., Evans, B. A., Falvo, M. R., Washburn, S., Superfine, R. Biomimetic cilia arrays generate simultaneous pumping and mixing regimes. *Proc. Natl. Acad. Sci. U. S. A.* **2010**, *107*, 15670-15675.
14. Gu, H., Boehler, Q., Cui, H., Secchi, E., Savorana, G., De Marco, C., Gervasoni, S., Peyron, Q., Huang, T., Pane, S., Hirt, A. M., Ahmed, D., Nelson, B. J. Magnetic cilia carpets with programmable metachronal waves. *Nat. Commun.* **2020**, *11*, 1-10.
15. Dong, X., Lum, G. Z., Hu, W., Zhang, R., Ren, Z., Onck, P. R., Sitti, M. Bioinspired cilia arrays with programmable nonreciprocal motion and metachronal coordination. *Sci. Adv.* **2020**, *6*, eabc9323.
16. Yang, X., Shang, W., Lu, H., Liu, Y., Yang, L., Tan, R., Wu, X., Shen, Y. An agglutinate magnetic spray transforms inanimate objects into millirobots for biomedical applications. *Sci. Robot.* **2020**, *5*, eabc8191.
17. Wu, Z. L., Buguin, A., Yang, H., Taulemesse, J. M., Le Moigne, N., Bergeret, A., Wang, X., Keller, P. Microstructured nematic liquid crystalline elastomer surfaces with switchable wetting properties. *Adv. Funct. Mater.* **2013**, *23*, 3070-3076.

18. Lee, S. G., Lee, D. Y., Lim, H. S., Lee, D. H., Lee, S., Cho, K. Switchable transparency and wetting of elastomeric smart windows. *Adv. Mater.* **2010**, *22*, 5013-5017.
19. Liu, Y., Mu, L., Liu, B., Kong, J. Controlled switchable surface. *Chem. Eur. J.* **2005**, *11*, 2622-2631.
20. Zhang, J., Ren, Z., Hu, W., Soon, R. H., Yasa, I. C., Liu, Z., Sitti, M. Voxelated three-dimensional miniature magnetic soft machines via multimaterial heterogeneous assembly. *Sci. Robot.* **2021**, *6*, eabf0112.

Response to Reviewer #2

The manuscript reports the results for the design of soft millimeter-sized robots by harnessing the wettability and reaction to external magnetic fields of ferrofluid droplets for controlled reconfigurability. The ferrofluid droplet at low wetting exhibits multimodal motions, such as stretching, jumping, spinning, tumbling, kayaking, and wobbling. Also, a single ferrofluid droplet can be controlled to split into multiple sub-droplets and then re-fuse back into a single droplet. The soft droplet can traverse various terrains and textures in unstructured environments. In addition, the ferrofluid droplets can be configured as a liquid capsule (at low wetting), enabling active liquid and solid cargo delivery, as a liquid cilia matrix (at high wetting), capable of pumping fluids, and as a liquid skin (at complete wetting), thereby facilitating the construction of multiple types of soft machine by adhesion to elastic materials.

The idea of using ferrofluids for the construction of miniature soft machines is very interesting, but not entirely new (see, e.g. Refs. [36-41]). As far as I can see, the main new aspect of the current manuscript is to also consider the wetting of various substrates by a ferrofluid droplet. This is an important aspect, which strongly increases the number of possibilities for the construction of miniature soft machines. Thus, this work improves the achievable complexity of small magnetic soft machines and boosts their future capabilities for robotics and biomedical engineering applications.

This is a "can do" manuscript, in which many kinds of manipulation of ferrofluids by time- and space-dependent magnetic field are demonstrated. This comes at the cost of often not very detailed explanations of the underlying mechanisms.

Response: We thank the reviewer for careful review and positive feedback on our work. We have answered every comment and made corresponding changes to the manuscript. Please check the following point-by-point response for details.

1. p.3: As explained above, the main new aspect of the current study is wettability. However, several parts of the manuscript also concern aspects which are not related to wetting. It would thus be good to explain in a bit more detail how the current study is distinguished from Refs. [36-41].

Response: We are grateful for the reviewer's comments. According to your suggestion, we have added the comparison in the “Introduction” part of the revised manuscript to highlight the differences between the previously reported literature and our work (**Please see section 1 and pages 3, 4 and 5**). Recent research works have demonstrated that ferrofluid droplets of different sizes act as various tiny machines^[1-6]. Fan et al. studied the centimeter-scale ferrofluid droplets' motion behavior under the magnetic field gradient forces^[1]. Sun et al. studied the micron-scale ferrofluid droplets' assembly behavior under magnetic torque^[2]. Yu et al. realized the transport and mixing of liquid samples in lab-on-a-chip by driving millimeter-scale ferrofluid droplets using magnetic field gradient forces generated by the electromagnetic navigation floor^[3]. Zhang et al. selectively modified the elastomeric surfaces via laser scanning and then penetrated them with ferrofluid to enable controllable deformation, folding, and functionality integration^[4]. Serwane et al. employed biocompatible, magnetically responsive ferrofluid microdroplets as local mechanical actuators to realize quantitative spatiotemporal measurements of mechanical properties *in vivo*^[5]. Zhou et al. used the ferrofluid as carriers for flexible electronic devices^[6]. Nonetheless, despite many advances in using ferrofluid to construct miniature soft machines, we can find significant differences between the previously

reported literature and our work. The related work mentioned by the reviewer focuses on the fluidity and paramagnetic properties of ferrofluids and ignores the dynamic behavior of ferrofluid droplets under different wetting conditions within the magnetic torque.

In contrast, this work is more focused on studying the dynamic wetting behavior of ferrofluid droplets under magnetic torque, exploiting their wetting properties to construct complex miniature soft machines, building bridges between fluidic materials and soft robotics. Such work builds on a broader range of efforts that draw inspiration from creatures in nature to develop a range of representative soft robots^[7]. To approach the advanced characteristics of natural organisms, these robotic systems are typically composed of materials with low Young's modulus or high deformability, such as polydimethylsiloxane^[8], shape memory alloy^[9], and dielectric elastomer^[10]. However, continued progress in developing soft robots requires further exploration of fluidic materials for actuation, mechanical adaptability, and intelligent control. Fluids are an essential component of biological creatures, permeating all aspects of our lives, and also play a crucial role in the energy transfer^[11], motion regulation^[12], etc., of soft robotic systems. Here we show how harnessing the wettability of ferrofluid droplets allows for controlled reconfigurability and the ability to create highly versatile fluidic soft machines. Our research provides a paradigm for integrating liquids into tiny soft-bodied machines.

Inspired by these aforementioned works, we demonstrate that versatile miniature soft machines can be constructed by exploiting the wetting properties and reconfigurability of ferrofluids. First, the wetting dynamics of ferromagnetic fluid droplets at different solid interfaces are studied. When the interaction between ferrofluid and substrate is weak (low wettability), the magnetic torque generated by the spatiotemporally programmed magnetic field

drives ferrofluid droplets to perform stretching, jumping, rotating, tumbling, kayaking, and wobbling motions. Due to their liquid properties, the ferrofluid droplets can split and fuse along the line or the plane in a controlled manner, and the number of fissions is highly controlled. The ferrofluid droplets are highly adaptable to both structured and unstructured environments by exploiting multimodal motion and controllable fission-fusion properties. The ferrofluid droplets can cross over successive obstacles, upstairs and through designated holes in walls in a jumping mode; navigate through highly curved small gaps and sharp turns in a stretching pattern; pass through narrow comb-like channels using fission and fusion. Moreover, the ferrofluid droplets can be reconfigured into miniature machines with multiple functions using wetting dynamics. At low wettability, the droplets can be reconfigured to serve as wireless liquid capsules for transporting liquid or solid cargo, which can travel through tortuous narrow channels to reach targeted positions and release the load-on-demand. When the interaction between ferrofluid droplets and the interface is strong (high wettability), the controllable deformability of the droplets allows them to act as arrays of liquid cilia, which are programmed to pump biological fluids. When the interaction between ferrofluid droplets and the interface is very strong (total wetting), the droplets can be reconfigured to serve as an active wireless liquid skin, which can controllably navigate near inanimate targets and then transform it into a soft machine through an adhesive strategy. Our proposed method to construct soft-bodied machines via the wettability of ferrofluid can inspire new construction strategies and achieve various unprecedented functionalities that could find broad applications in biomedical engineering.

We have added new results in the revised manuscript and supporting information to highlight the innovation.

2. p.3: When the authors mention "low wettability", it is not obvious of what: the cargo? a substrate?

Response: We are grateful for the reviewer's expert advice. To avoid ambiguity, we indicate whether it is with the substrate or with the cargo when we refer to low wettability. **For more details, please see pages 3 and 4 of the revised manuscript.**

3. p.3: The authors state that "In-depth studies have NOT yet been carried out to characterize the wetting dynamics of ferrofluid droplets ...". I find this a bit surprising, after many years of intensive studies of ferrofluids. In fact, the authors themselves cite Refs. [44-48] -- for not obvious reasons these are only mentioned in the Methods section! In addition there are, e.g. the publications: S.Shyam et al., Coll. Surf. A 586, 124116 (2020); V.A. Roodan et al., Soft Matter, 16, 9506-9518 (2020).

Response: We are grateful for the reviewer's comments. Our claim was an incorrect overstatement and we thank the reviewer for the kind reminder. The relevant documents mentioned by the reviewer and what we can currently find have been added and acknowledged in the revised manuscript (**please see references 49 and 50**) because of the authors' significant contributions in this field^[13,14]. To avoid being misleading, the description "In-depth studies have not yet been carried out to characterize the wetting dynamics of ferrofluid droplets, their behavior in coupling with different interfaces, and the individual dynamics under magnetic torque; thus, it is challenging to realize their full potential be reconfigured as miniature soft machines" has been modified in the revised manuscript as follows: "The dynamic behavior of ferrofluid droplets in wetting with different interfaces is underutilized, and the individual dynamics under

magnetic torque have not been adequately investigated; thus, it is challenging to realize their full potential to be reconfigured as miniature soft machines" (please see section 1 and page 3).

4. p.3: The authors claim that "The ferrofluid droplets are highly adaptable to both structured and unstructured environments ...". These are many claims! Are all of them substantiated in the manuscript?

Response: Many thanks to the reviewer for the comments. The high adaptability of ferrofluid droplets to structured and unstructured environments has been substantiated in section 2.3. (Environmental adaptability) of the manuscript. A structured environment refers to an environment with a uniform surface and regular and stable changes in structure and size^[15]. Unstructured environment refers to terrain with uneven surface material properties and irregular and unstable changes in structure and size, usually the natural environment^[16]. In section 2.3., we try to select common and representative terrains or environments to verify the environmental adaptability of the ferrofluid droplets on low wettability. For example, when investigating the motion behaviors independently, we artificially customized these structured environments, such as successive obstacles, upstairs, designated holes in walls, small curved gaps, sharp turns, and narrow comb-like channels. Ferrofluid droplet-based robots are well adapted to the terrains. For the potential biomedical application scenario, we directly select the pig bladder surface, the pig stomach's inner wall, and the human placental vessels as the unstructured environment to demonstrate the environmental adaptability of multimodal motion of a ferrofluid droplet-based robot. To elaborate more specifically, the description "The ferrofluid droplets are highly adaptable to both structured and unstructured environments by exploiting multimodal motion and controllable

fission-fusion properties" has been modified as follows "The ferrofluid droplet-based robots are highly adaptable to both artificial and biological environments by exploiting multimodal motion and controllable fission-fusion properties."
For more details, please see page 4 of the revised manuscript.

5. p.4: What are "wireless" liquid capsules?

Response: We are grateful for the reviewer's comments. Wireless (untethered) means electrical or pneumatic tethers do not limit miniature robots^[17,18]. Traditional wired (tethered) miniature robots often require a tethered connection to support pneumatic or electrical hardware^[17]. Conventional capsules are also wireless, and the emphasis on wireless here is redundant. So, we modified "wireless liquid capsule" as "liquid capsule" in the revised manuscript. **For more details, please see page 15 of the revised manuscript.**

6. p.6: Wetting properties depend on three surface tensions! What is the "suspending" phase? I guess water. Furthermore, it seems to me that as the carrier fluid of a ferrofluid is "oily", the wetting properties should be mainly determined by the hydrophobicity of the substrate.

Response: Many thanks to the reviewer for valuable comments. We agree with the reviewer that wetting properties depend on surface tensions. And indeed, as the reviewer stated, since the carrier fluid of the ferrofluid is "oily", the hydrophobicity of the substrate determines the wetting properties. We select substrates of common materials and characterize the contact angles of ferrofluid droplets on different substrates with and without a magnetic field in the aqueous phase. The suspending phase used in our work is water, which is clearly described in Figure 2A. In the revised version, the water phase is emphasized

again in the text and legend to avoid being misunderstanding. **For more details, please see page 6 of the revised manuscript.**

7. p. 11: No wetting seems to be involved in Sec. 2.3.

Response: We are grateful for the reviewer's valuable comments. In Section 2.3, we mainly highlighted the adaptability of ferrofluid droplet-based robots to artificial and biological environments, driven by magnetic torque. The environmental adaptability of ferrofluid droplets is crucial for their use as mobile liquid capsules, liquid cilia, and liquid skin. In addition, this environmental adaptability of ferrofluid droplet-based robots can only occur in the non-wetting state. The substrate used to build the structured environment is hydrogel material. The non-adhesion of the ferrofluid to the hydrogel allows the ferrofluid droplets to navigate through complex artificial environments, such as successive obstacles, upstairs, small curved gaps, and so on. To avoid misunderstandings, we have highlighted the factor of non-adhesion to the surrounding substrate in the revised manuscript, indicating that this environmental adaptability is only possible in the non-wetting condition. **For more details, please see pages 11, 12, 13, and 14 of the revised manuscript.**

8. p. 11: Medical robots: Are ferrofluids sufficiently biocompatible? Where do they operate? Clearly the droplets would be too large for blood vessels.

Response: Many thanks to the reviewer for the comments. Many thanks to the reviewer for the comments. According to your suggestion, we have performed *in vitro* cytotoxicity of ferrofluid in the revised manuscript (**please see page 14 and Figure S17**). The biocompatibility of ferrofluids has been demonstrated in several previous studies^[19-21], and ferrofluids are widely used in biomedical

fields^[22,23]. To demonstrate the biocompatibility of our ferrofluids, we performed in vitro cytotoxicity. NIH 3T3 cells with a density of 2000 cells/well were seeded in a 96-well plate, followed by 12 h incubation in 100 μ L Eagle's Minimum Essential Medium with 10% fetal bovine serum. A ferrofluid suspension was obtained by sonicating 1 mg of ferrofluid into 1 ml of fresh medium. Then the medium was discarded, and 100 μ L fresh medium containing different concentrations of ferrofluid suspension was added to the NIH 3T3 cells. Subsequently, these different samples were cocultured with the cells for 48 h. The MTS assay quantified cell viability. 10 μ L MTS solution was added to each well, followed by another 2 h incubation. Then nanoparticles were concentrated on the bottom with a permanent magnet, and the supernatant solution was transferred to a new 96-well plate. The absorbance was detected at 490 nm with a microplate reader. All of the tests were repeated three times. As shown in **Figure R2.1**, with a concentration up to 800 μ g/mL, the ferrofluid was nontoxic to the NIH 3T3 cells, indicating their biocompatibility.

Figure R2.1. NIH 3T3 cells viabilities after 48 h incubation with ferrofluid suspension with different concentrations.

Furthermore, we agree with the reviewer that millimeter-scale ferrofluid droplets are unsuitable for direct application to blood vessels. Human placental vessels were chosen in our work to demonstrate environmental adaptability because of their variable inner diameter. Experimental results verify the adaptability of ferrofluid droplets to changing cavities. The ideal environment for ferrofluids to operate is in narrow lumens, such as the bile ducts in the human body. Traditional flexible catheters do not easily reach the interior of the bile duct, whereas ferrofluid droplets can non-invasively travel through the narrow bile duct to the target site and perform targeted functions, such as drug release.

9. p.12: The authors state that "... inside an annular channel, the ferrofluid droplet constantly stretches and shrinks to achieve locomotion by utilizing the frictional force of the sidewall." I don't understand this argument. What breaks the symmetry between clockwise and counter-clockwise motion?

Response: We thank the reviewer for the comments. According to your suggestion, we have clarified the motion mechanism of ferrofluid droplets inside the annular channel in the revised manuscript (**please see page 13 and Figure S16**). When the ferrofluid droplet navigates through the annular channel and constantly stretches and shrinks, it also continually changes direction with the magnetic field. As shown in **Figure R2.2**, the ferrofluid droplet begins to elongate and simultaneously touch both sides of the wall under the magnetic field at stage 1. In stage 2, the elongated ferrofluid droplet begins to contract. At this time, the motion of the ferrofluid is symmetrical, and there is no net displacement. Then, the angle between the external oscillating magnetic field and the horizontal axis is increased along the clockwise direction. In stage 3, the ferrofluid rotates and elongates following the external magnetic field, and one

end of the ferrofluid droplet will touch the right-side wall first and then be supported by the right-side wall to elongate to the left. At this time, the motion symmetry is broken. In stage 4, the elongated ferrofluid droplet begins to contract again. Finally, the ferrofluid droplet moves forward a small distance compared to its initial position. The counterclockwise movement of the ferrofluid adopts a similar strategy. To clarify the motion mechanism of ferrofluid droplets inside the annular channel, the description "When moving inside an annular channel with an inner diameter of 2 mm, the ferrofluid droplet constantly stretches and shrinks to achieve locomotion by utilizing the frictional force of the sidewall." has been modified as follows: "When moving inside an annular channel with an inner diameter of 2 mm, the ferrofluid droplets are constantly turning as they stretch and contract to achieve locomotion by utilizing the supporting force of the sidewall" (please see section 2.3 and page 13).

Figure R2.2. Mechanism for clockwise and counterclockwise movement of ferrofluid droplets in an annular channel. The black arrows indicate the

direction of motion of the ferrofluid droplets and the red arrows indicate the direction of the magnetic field.

10. p.14: The switching between different modes of operation in a heterogeneous environment requires detailed knowledge about instantaneous location and environment! Can this be achieved in medical applications?

Response: We are grateful for the reviewer's valuable comments. We agree with the reviewer that transitioning between different operating modes can be adapted to complex unstructured environments but requires detailed knowledge of the transient position and environment. Theoretically, it could also be achieved in biomedical applications. X-ray fluoroscopy could offer reliable real-time imaging for state observation of the ferrofluid droplet-based robot being manipulated inside the human body because a ferrofluid droplet will be naturally visible under X-ray due to radiopaque magnetic particles. Another critical point is that externally applied magnetic fields do not cause interference in X-ray imaging, which is why commercialized magnetic manipulation systems have been used along with a C-arm fluoroscope for real-time X-ray imaging. However, the fact that X-ray fluoroscopy provides only planar 2D projection imaging at a time implies that an integrated imaging and actuation platform will be needed to facilitate flexible and dexterous manipulation of soft magnetic robots in 3D environments. The C-arm fluoroscopy system has evolved into a multi-DOF robotic platform capable of automated rotation and angulation with flexible maneuverability for complex image-guided interventions^[24]. In this regard, we envision that next-generation magnetic manipulation platforms will have an integrated X-ray fluoroscopy imaging system that can be synchronized with magnetic actuation. This integrated magnetic manipulation platform can provide real-time feedback for miniature robot guidance.

11. p.14: Syringe injection of a cargo liquid: Seems to require complete wetting by the ferrofluid to avoid instability and ejection!

Response: We thank the reviewer for the comments. The injection of liquid cargo requires complete wetting between the syringe needle and the ferrofluid droplet (**Figure R2.3**). To avoid instability and ejection, a typical 5 μL droplet is injected with a liquid load of approximately 0.5 μL . **The relevant Figure has been added to the revised version, and please see page 15 of the revised manuscript for more details.**

Figure R2.3. The syringe needle is completely infiltrated with ferrofluid.

12. p.19: Are the anchoring sites of the cilia little dimples in the substrate? Otherwise, why would the ferrofluid droplets not move or be displaced on the substrate?

Response: Many thanks to the reviewer for the comments. The anchoring sites of the cilia are little dimples on the substrate. Because of the little dimples, the ferrofluid droplets do not move on the substrate. This has been described in detail in the original manuscript, and we have included the following "As shown in Figure 6B, at $t = 0$ s, the ferrofluid droplet remains elliptical and is confined within a 2 mm semicircular pit in the substrate (**Video S7**) " for your reference.

13. p.18: I don't agree with the statement "Unlike natural or other artificial cilia, this presented liquid cilia based on ferrofluid exhibit nonreciprocal motion, even though they do not have a power stroke or a recovery stroke." Of course, both types of strokes are present.

Response: We are grateful for the reviewer's valuable comments. We agree with the reviewer that the liquid cilia also have a power stroke and a recovery stroke during their motion. The elongation-rotation process of the ferrofluid cilia is the power stroke. The recovery stroke occurs when the liquid cilia contracts. Unlike natural cilia or other artificial cilia made from elastomer, which are based on buckling motion, liquid cilia rely on their elongation and contraction motion. To avoid being misleading, the description "Unlike natural or other artificial cilia, this presented liquid cilia based on ferrofluid exhibit nonreciprocal motion, even though they do not have a power stroke or a recovery stroke." has been modified as follows "Unlike natural cilia or other artificial cilia made from elastomer depending on bucking motion²⁷, this presented liquid cilia based on elongation and contraction exhibit nonreciprocal motion with periodic power stroke and recovery stroke." **For more details, please see page 19 of the revised manuscript.**

14. p.20: How effective are the cilia arrays in pumping fluid? How does the fluid velocity of 3.9 mm/s relate to the periodicity of the array and the beat frequency? How does this compare to other artificial cilia?

Response: We thank the reviewer for the comments. According to your suggestion, we have investigated the relationship between the fluid velocity and the periodicity and beat frequency of liquid cilia, and compared the performance of liquid cilia with other artificial cilia (**please see page 21, Figure S26 and Table S1**). The pumping speed of 3.9 mm/s achieved by the ciliary array is when the suspended phase is water (Reynolds number > 1). However, pumping viscous fluids at a low Reynolds number (Re ; ~ 0.001 to 0.01) is challenging because of the no-slip boundary conditions but is essential in many biological systems, such as tubal transportation of ova in female fallopian tubes, where ciliary motion contributes to efficient transportation in addition to muscular contractility^[31]. Here, we used glycerol to create a low Reynolds number environment and investigated the relationship between the velocity of the pumped fluid and the periodicity and the beat frequency of the array. As shown in **Figure R2.4a**, we demonstrate that the array of 7 cilia can efficiently transport tracer particles in a glycerol environment (The tracer particles is hydrogel sphere with a diameter of about 2 μm , $B = 9$ mT, $f = 0.5$ Hz). When the spacing between the liquid cilia was changed from 2 μm to 6 μm ($B = 9$ mT, $f = 0.8$ Hz), the fluid velocity first increases and then decreases, and the fluid rate is the largest at 4 μm spacing, about 0.15 mm/s (**Figure R2.4b**). The relationship between the fluid velocity of the liquid cilia array and the beat frequency is shown in **Figure R2.4c**. The fluid rate gradually increases as the frequency increases and then gradually decreases. When the frequency is 1.1 Hz, the pumping velocity of the liquid ciliary array is about 0.195 mm/s. In addition, liquid cilia have good pumping efficiency compared to other artificial cilia at low Re (**Table R2.1**).

Figure R2.4. Liquid cilia array for pumping viscous fluids. a. Viscous fluids (glycerol) pumping by the array of 7 liquid cilia visualized by particle transportation. Scale bar: 2 mm. **b.** The pumping velocity versus the periodicity of the cilia array. **c.** The pumping velocity versus the beat frequency of the liquid array.

References	Normalized velocity	Fluid velocity	Driving frequency	Cilia length	Cilia number	Materials
	v/L	v	f	L	N	
25	0.1	3.3 $\mu\text{m/s}$	1 Hz	31 μm	3×3	Superparamagnetic microparticles
26	0.143	500 $\mu\text{m/s}$	7 Hz	500 μm	1×6	Polydimethylsiloxane

27	0.0214	75 $\mu\text{m/s}$	10 Hz	350 μm	3×3	Polydimethylsiloxane
28	0.011	9 $\mu\text{m/s}$	34 Hz	25 μm	3000	Silicone elastomer
29	0.25	83 $\mu\text{m/s}$	0.083 Hz	4 mm	8×8	Silicone elastomer
30	0.38	0.95 mm/s	2.5 Hz	1 mm	6×6	Silicone elastomer
This work	0.147	0.195 mm/s	1.1 Hz	1.2 mm	1×7	Ferrofluid

Table R2.1. Comparison of the fluid pumping performance using artificial cilia with literature. The normalized velocity, shows how fast the flow it can generate by one beating cycle.

15. p.21: The ferrofluid penetrates the body of the elastomer sheet -- I would not really denote this as wetting!

Response: We thank the reviewer for the comments. According to your suggestion, we have corrected the original statement to avoid being misleading (please see section 2.6 and page 22). We agree with the reviewer that when ferrofluids penetrate the body of an elastomer sheet, it cannot be called wetting. Before the ferrofluid droplet penetrates the elastomer sheet, the ferrofluid droplet is first totally wetted onto the surface of the elastomer sheet under the control of a magnetic field. So, the ferrofluid droplet acts as a wireless liquid skin, and the critical process by which a series of miniature soft robots can be built through an adhesion strategy is sequentially divided into three stages. In the first stage, the ferrofluid droplet is driven by the magnetic field to reach the target elastomer and then, under the influence of the magnetic field, thoroughly wets the elastomer surface. In the second stage, ferrofluid penetrates the silicone elastomer surface layer. Due to the high mutual solubility of the hydrocarbon solvent and the elastomeric sheet, the solvent diffuses rapidly from the surface

to the inner layer along its concentration gradient. As a result, the upper part of the elastomer sheet becomes swollen and inflated. At the same time, most dispersed nanoscale iron oxide particles concentrate and precipitate out, and some adhere to the surface. In the third stage, ferrofluid penetrates the body of the elastomer sheet. The solvent continues to evaporate, and the magnetic particles start to agglomerate. Iron oxide nanoparticles aggregate on the scanned surface and form a continuous rigid layer.

16. p.22: How is the spider structure (with four legs) obtained from a uniform coverage?

Response: We are grateful for the reviewer's comments. The spider's structure is obtained due to the deformation caused by the solvent's penetration into the elastomer's surface layer. Because of the high intermiscibility of the hydrocarbon solvent and elastomeric sheet, the solvent rapidly diffuses along its concentration gradient from the surface layer to the inner layers. Consequently, the upper part of the elastomeric sheet becomes swollen and expands. Simultaneously, most dispersed nanoscale iron oxide particles concentrate and precipitate out, and some adhere to the surface because of the rapid loss of solvent. The elastomeric sheet is observed to bunch up during this stage. The iron oxide nanoparticles become agglomerated on the surface, producing a continuous stiff layer. The 3D shape is thereby immobilized and does not notably change afterward.

17. p.22: "... fish robot that swims under the oscillating magnetic field"? The propulsion mechanism is not explained! Is this a scallop-type propulsion at finite Reynolds number?

Response: We are grateful for the reviewer's valuable comments. We apologize for not clearly describing the propulsion mechanism. According to your suggestion, we have explained the propulsion mechanism of fish robot in the revised manuscript (**please see section 2.6, page 23 and Figure S32**). We have now added discussion regarding the realization of the biomimetic swimming gaits and the propulsion mechanism of the fish robot. The propulsion mechanism for the fish robot is the same as the body-caudal fin propulsion swimming robot^[32,33]. The head of the fish robot is firstly wetted by the ferrofluid droplet, which enables it to be controlled by the external magnetic field. The oscillating magnetic field will force the robot's head to sway from side to side, a process similar to excitation at one terminal of a beam at its first-order bending vibration frequency, thus causing the beam to bend. Since the head is heavier and more rigid (since the elastomer sheet at this region is wetted by the ferrofluid droplet), the tail displacement will be more pronounced. Then, when the tail swings from side to side, the water on the rear side is continuously pushed away from the body, causing vortices behind its tail (as the COMSOL Multiphysics simulation result in **Figure R2.5** suggests) and the reaction force generated by the fluid on the robot's body during this process will eventually push the robot forward. The relationship between the deformation of the fish robot body and the fluid force is given by

$$\rho_s \ddot{\mathbf{u}}_s = \text{div } \mathbf{S} + \mathbf{f}_v \quad (1)$$

where ρ_s is the mass density of the fish robot, \mathbf{u}_s is the displacement vector, \mathbf{S} is the reference stress applied to the fish robot, \mathbf{f}_v is the fluid force exerted on the solid structure. The fluid is governed by the forces balance and mass conservation equations as follows:

$$\rho_f \dot{\mathbf{v}}_f + \rho_f (\nabla \mathbf{v}_f) \mathbf{v}_f = \text{div } \Gamma + \mathbf{f} \quad (2)$$

$$\dot{\rho}_f + \text{div}(\rho_f \mathbf{v}_f) = 0 \quad (3)$$

Where ρ_f is the mass density of the fluid, \mathbf{v}_f is the fluid spatial velocity, \mathbf{f} is the force of solid acting on the fluid, and the stress Γ is given by:

$$\Gamma = -p\mathbf{I} + 2\mu_f(\text{sym} \nabla \mathbf{v}_f) - \frac{2}{3}\mu_f(\text{div} \mathbf{v}_f)\mathbf{I} \quad (4)$$

where p is the fluid pressure, and μ_f is the dynamic viscosity

Figure R2.5. The experimental results and simulation results of the gait of the ‘fish robot’.

References

1. Fan, X., Dong, X., Karacakol, A., Xie, H., Sitti, M. Reconfigurable multifunctional ferrofluid droplet robots. *Proc. Natl. Acad. Sci. U. S. A.* **2020**, *117*, 27916-27926.
2. Sun, M., Fan, X., Tian, C., Yang, M., Sun, L., Xie, H. Swarming microdroplets to a dexterous micromanipulator. *Adv. Funct. Mater.* **2021**, *31*, 2011193.
3. Yu, W., Lin, H., Wang, Y., He, X., Chen, N., Sun, K., Lo, D., Cheng, B., Yeung, C., Tan, J., Carlo, D., Emaminejad, S. A ferrobatic system for automated microfluidic logistics. *Sci. Robot.* **2020**, *5*, eaba4411.
4. Zhang, S., Ke, X., Jiang, Q., Ding, H., Wu, Z. Programmable and reprocessable multifunctional elastomeric sheets for soft origami robots. *Sci. Robot.* **2021**, *6*, eabd6107.
5. Serwane, F., Mongera, A., Rowghanian, P., Kealhofer, D. A., Lucio, A. A., Hockenbery, Z. M., Campas, O. In vivo quantification of spatially varying mechanical properties in developing tissues. *Nat. Methods* **2017**, *14*, 181-186.
6. Zhou, M., Wu, Z., Zhao, Y., Yang, Q., Ling, W., Li, Y., Xu, H., Wang, C., Huang, X. Droplets as carriers for flexible electronic devices. *Adv. Sci.* **2019**, *6*, 1901862.
7. Rus, D., Tolley, M. T. Design, fabrication and control of soft robots. *Nature* **2015**, *521*, 467-475.
8. Hu, W., Lum, G. Z., Mastrangeli, M., Sitti, M. Small-scale soft-bodied robot with multimodal locomotion. *Nature* **2018**, *554*, 81-85.

9. Laschi, C., Cianchetti, M., Mazzolai, B., Margheri, L., Follador, M., Dario, P. Soft robot arm inspired by the octopus. *Adv. Robot.* **2012**, *26*, 709-727.
10. Chen, Y., Zhao, H., Mao, J., Chirarattananon, P., Helbling, E. F., Hyun, N. S. P., Clarke, D., Wood, R. J. Controlled flight of a microrobot powered by soft artificial muscles. *Nature* **2019**, *575*, 324-329.
11. Aubin, C. A., Choudhury, S., Jerch, R., Archer, L. A., Pikul, J. H., Shepherd, R. F. Electrolytic vascular systems for energy-dense robots. *Nature* **2019**, *571*, 51-57.
12. Wehner, M., Truby, R. L., Fitzgerald, D. J., Mosadegh, B., Whitesides, G. M., Lewis, J. A., Wood, R. J. An integrated design and fabrication strategy for entirely soft, autonomous robots. *Nature* **2016**, *536*, 451-455.
13. Shyam, S., Asfer, M., Mehta, B., Mondal, P. K., Almutairi, Z. A. Magnetic field driven actuation of sessile ferrofluid droplets in the presence of a time dependent magnetic field. *Colloids Surf. A Physicochem. Eng. Asp.* **2020**, *586*, 124116.
14. Roodan, V. A., Gómez-Pastora, J., Karampelas, I. H., González-Fernández, C., Bringas, E., Ortiz, I., Chalmers, J. J., Furlani, E. P., Swihart, M. T. Formation and manipulation of ferrofluid droplets with magnetic fields in a microdevice: a numerical parametric study. *Soft Matter* **2020**, *16*, 9506-9518.
15. Caley, J. A., Lawrance, N. R., Hollinger, G. A. Deep learning of structured environments for robot search. *Auton. Robot.* **2019**, *43*, 1695-1714.
16. Guastella, D. C., Muscato, G. Learning-based methods of perception and navigation for ground vehicles in unstructured environments: A review. *Sensors* **2020**, *21*, 73.

17. Rich, S. I., Wood, R. J., Majidi, C. Untethered soft robotics. *Nat. Electron.* **2018**, *1*, 102-112.
18. Sitti, M., Ceylan, H., Hu, W., Giltinan, J., Turan, M., Yim, S., Diller, E. Biomedical applications of untethered mobile milli/microrobots. *Proc. IEEE* **2015**, *103*, 205-224.
19. Kose, A. R., Fischer, B., Mao, L., Koser, H. Label-free cellular manipulation and sorting via biocompatible ferrofluids. *Proc. Natl. Acad. Sci. U. S. A.* **2009**, *106*, 21478-21483.
20. Arana, M., Bercoff, P. G., Jacobo, S. E., Zélis, P. M., Pasquevich, G. A. Mechanochemical synthesis of MnZn ferrite nanoparticles suitable for biocompatible ferrofluids. *Ceram. Int.* **2016**, *42*, 1545-1551.
21. Khoramian, S., Saeidifar, M., Zamanian, A., Saboury, A. A. Synthesis and characterization of biocompatible ferrofluid based on magnetite nanoparticles and its effect on immunoglobulin G as an immune protein. *J. Mol. Liq.* **2019**, *273*, 326-338.
22. Kole, M., & Khandekar, S. Engineering applications of ferrofluids: A review. *J. Magn. Magn. Mater.* **2021**, *537*, 168222.
23. Socoliuc, V., Avdeev, M. V., Kuncser, V., Turcu, R., Tombácz, E., Vekas, L. Ferrofluids and bio-ferrofluids: looking back and stepping forward. *Nanoscale* **2022**, *14*, 4786-4886.
24. Kim, Y., Zhao, X. Magnetic soft materials and robots. *Chem. Rev.* **2022**, *122*, 5317-5364.

25. Vilfan, M., Potočnik, A., Kavčič, B., Osterman, N., Poberaj, I., Vilfan, A., Babič, D. Self-assembled artificial cilia. *Proc. Natl. Acad. Sci. U. S. A.* **2010**, *107*, 1844-1847.
26. Rockenbach, A., Schnakenberg, U. The influence of flap inclination angle on fluid transport at ciliated walls. *J. Micromechanics Microengineering* **2017**, *27*, 015007.
27. Zhang, S., Wang, Y., Lavrijsen, R., Onck, P. R., den Toonder, J. M. Versatile microfluidic flow generated by moulded magnetic artificial cilia. *Sensors Actuators B Chem.* **2018**, *263*, 614-624.
28. Shields, A. R., Fiser, B. L., Evans, B. A., Falvo, M. R., Washburn, S., Superfine, R. Biomimetic cilia arrays generate simultaneous pumping and mixing regimes. *Proc. Natl. Acad. Sci. U. S. A.* **2010**, *107*, 15670-15675.
29. Gu, H., Boehler, Q., Cui, H., Secchi, E., Savorana, G., De Marco, C., Gervasoni, S., Peyron, Q., Huang, T., Pane, S., Hirt, A. M., Ahmed, D., Nelson, B. J. Magnetic cilia carpets with programmable metachronal waves. *Nat. Commun.* **2020**, *11*, 1-10.
30. Dong, X., Lum, G. Z., Hu, W., Zhang, R., Ren, Z., Onck, P. R., Sitti, M. Bioinspired cilia arrays with programmable nonreciprocal motion and metachronal coordination. *Sci. Adv.* **2020**, *6*, eabc9323.
31. Lyons, R. A., Saridogan, E., Djahanbakhch, O. The reproductive significance of human Fallopian tube cilia. *Hum. Reprod. Update* **2006**, *12*, 363-372.
32. Lee, K. Y., Park, S. J., Matthews, D. G., Kim, S. L., Marquez, C. A., Zimmerman, J. F., Ardoña, H. A. M., Kleber, A. G., Lauder, G. V., Parker,

K. K. An autonomously swimming biohybrid fish designed with human cardiac biophysics. *Science* **2022**, 375, 639-647.

33. Borazjani, I., Sotiropoulos, F. On the role of form and kinematics on the hydrodynamics of self-propelled body/caudal fin swimming. *J. Exp. Biol.* **2010**, 213, 89-107.

Response to Reviewer #3

This manuscript demonstrates the versatility of ferrofluid droplets for robotic applications. The authors present interesting applications that could be of relevance to soft robots. I believe this manuscript would be of interest to the readership of the Nature Communications and would be happy to recommend it for publications; I only have a few minor comments/questions:

Response: We thank the reviewer for careful review and high evaluation of this work. Your constructive comments and suggestions have helped us to further improve our manuscript.

1. For the biomedical applications, can the authors comment on a) biocompatibility of the ferrofluids, and whether it would be ok for the fluids to spread through the tissues, etc, and b) whether the use of a magnet given the required distance and imaging, etc is a realistic way of guiding ferrofluids in medical applications?

Response: Many thanks to the reviewer for the comments. Many thanks to the reviewer for the comments. According to your suggestion, we have performed *in vitro* cytotoxicity of ferrofluid in the revised manuscript (**please see page 14 and Figure S17**). The biocompatibility of ferrofluids has been demonstrated in several previous studies^[1-3], and ferrofluids are widely used in biomedical fields^[4,5]. To demonstrate the biocompatibility of our ferrofluids, we performed *in vitro* cytotoxicity. NIH 3T3 cells with a density of 2000 cells/well were seeded in a 96-well plate, followed by 12 h incubation in 100 μ L Eagle's Minimum Essential Medium with 10% fetal bovine serum. A ferrofluid suspension was obtained by sonicating 1 mg of ferrofluid into 1 ml of fresh

medium. Then the medium was discarded, and 100 μL fresh medium containing different concentrations of ferrofluid suspension was added to the NIH 3T3 cells. Subsequently, these different samples were cocultured with the cells for 48 h. The MTS assay quantified cell viability. 10 μL MTS solution was added to each well, followed by another 2 h incubation. Then nanoparticles were concentrated on the bottom with a permanent magnet, and the supernatant solution was transferred to a new 96-well plate. The absorbance was detected at 490 nm with a microplate reader. All of the tests were repeated three times. As shown in **Figure R3.1**, with a concentration up to 800 $\mu\text{g}/\text{mL}$, the ferrofluid was nontoxic to the NIH 3T3 cells, indicating their biocompatibility.

As stated by the reviewer, using permanent magnets in combination with imaging devices is also a way to guide the motion of ferrofluid droplets in medical applications. It should be possible to meet some medical application scenarios that do not require high levels of motion control. The direct use of permanent magnets also has several limitations. Due to the large magnetic field gradient forces of permanent magnets, ferrofluid droplets can easily split. In addition, permanent magnets are limited in the type of magnetic field they can generate; for example, they cannot generate conical magnetic fields. This moves ferrofluid droplets relatively simply, reducing their adaptability to the complex environment in the body. While orthogonally placed coil systems can provide multiple magnetic field types and give various modes of motion for ferrofluid droplet actuation, their limited working space further limits the practical application of ferrofluid droplets or miniature robots of the same size. To date, various configurations of magnetic drive systems have been developed that have a larger working space and retain the programmable property of the magnetic field^[6]. Integrating imaging devices with these magnetic manipulation platforms holds promise for ferrofluid biomedical applications. We will investigate the application of ferrofluids in biomedical fields in depth in future work.

Figure R3.1. NIH 3T3 cells viabilities after 48 h incubation with ferrofluid suspension with different concentrations.

2. In Fig. 3, why do the drops coalesce?

Response: We are grateful for the reviewer's comments. According to your suggestion, we have added the reason for ferrofluid droplets coalescence in the revised manuscript (**please see section 2.2 and page 11**). The ferrofluid droplets begin to approach each other due to the magnetic attraction under the magnetic field with low frequency. As the droplets proceed into close contact with each other, their interfaces flatten, and the fluid film between the droplets starts to drain. Eventually, when the fluid film becomes thin enough, the van der Waals intermolecular forces dominate and cause the fluid film to rupture, which ultimately contributes to droplet coalescence^[7-9].

3. Can you elaborate on the mechanism of drug release in Fig. 5? If the contraction of the liquid droplet is causing the release, how come this release is not expected during the locomotion stages before the droplet gets to the target?

Response: We thank the reviewer for the comments. According to your suggestion, we have elaborated on the mechanism of drug release in detail in the revised manuscript (**please see page 16 and Figure S19**). The liquid capsule releases the drug by which a vertical magnetic field induces its elongation, resulting in a thinning of the ferrofluid film surrounding the liquid cargo, which then releases it. There are two critical factors in the cargo release process from liquid capsules: the first is the applied magnetic field has the maximum value of strength, and the second is that the liquid capsule is in a vertical state. The injected drug will remain above the capsule due to its density. Under the vertical magnetic field, the liquid capsule is in a standing state, and the injected cargo is located at the top of the tip of the capsule. As the strength of the vertical magnetic field continues to increase, the ferrofluid film surrounding the liquid cargo is continuously thinned. Finally, the film ruptures to release the liquid load (**Figure R3.2**). However, during the experiment, since our system reached the maximum magnetic field strength of 9 mT, the deformation of the ferrofluid was limited, and sometimes the liquid cargo could not be directly discharged. At this time, by applying a vertical oscillating magnetic field, the ferrofluid droplets are continuously extended and contracted. The ferrofluid droplet is significantly deformed under the action of inertial force, and then the membrane is ruptured to release the liquid cargo (**Figure R3.2**).

As for how to ensure that the liquid cargo does not leak during the movement, there are mainly the following reasons. Firstly, when driving the liquid capsule to move, the magnetic field strength we apply is only half of the maximum magnetic field strength (about 4 mT), which causes the ferrofluid film around the liquid cargo not too thin. Secondly, the movement process generally adopts the stretching mode or the tumbling mode. When the stretching mode is used, the pitch angle is generally less than 10° , and the liquid cargo is basically

unchanged above the liquid capsule (**Figure R3.2**). When moving in a tumbling mode, the liquid cargo also remains in position relative to the capsule and does not induce rupture (**Figure R3.2**).

Figure R3.2. Mechanisms of drug release and mechanisms to maintain non-leakage during motion.

4. Related to liquid cilia in Fig. 6, qualitative arguments related to Scallop theorem are mentioned; a rough estimate of Reynolds number, however, shows that the flow is likely at $Re > 1$, and these arguments may not be relevant.

Response: We are grateful for the reviewer’s expert advice. To avoid being misleading, the description "According to the scallop theorem, this nonreciprocal motion is essential for the cilia to pump the liquid in their vicinity at low Reynolds numbers²⁷. Unlike natural or other artificial cilia, this presented

liquid cilia based on ferrofluid exhibit nonreciprocal motion, even though they do not have a power stroke or a recovery stroke." has been modified as follows "Unlike natural or other artificial cilia depending on bucking-based motion, this presented liquid cilia based on elongation and contraction exhibit nonreciprocal motion." (please see section 2.5 and page 19).

In addition, we used glycerol to create a low Reynolds number environment and investigated the relationship between the velocity of the pumped fluid and the periodicity and the beat frequency of the array, and compared the performance of liquid cilia with other artificial cilia (please see page 21, Figure S26 and Table S1). As shown in Figure R3.3a, we demonstrate that the array of 7 cilia can efficiently transport tracer particles in a glycerol environment (The tracer particles is hydrogel sphere with a diameter of about 2 μm , $B_m = 9 \text{ mT}$, $f = 0.5 \text{ Hz}$). When the spacing between the liquid cilia was changed from 2 μm to 6 μm ($B_m = 9 \text{ mT}$, $f = 0.8 \text{ Hz}$), the fluid velocity first increases and then decreases, and the fluid rate is the largest at 4 μm spacing, about 0.15 mm/s (Figure R3.3b). The relationship between the fluid velocity of the liquid cilia array and the beat frequency is shown in Figure R3.3c. The fluid rate gradually increases as the frequency increases and then gradually decreases. When the frequency is 1.1 Hz , the pumping velocity of the liquid ciliary array is about 0.195 mm/s . In addition, liquid cilia have good pumping efficiency compared to other artificial cilia at low Re (Table R3.1).

Figure R3.3. Liquid cilia array for pumping viscous fluids. a. Viscous fluids (glycerol) pumping by the array of 7 liquid cilia visualized by particle transportation. Scale bar: 2 mm. b, The pumping velocity versus the periodicity of the cilia array. c. The pumping velocity versus the beat frequency of the liquid array.

References	Normalized velocity	Fluid velocity	Driving frequency	Cilia length	Cilia number	Materials
	v/fL	v	f	L	N	
25	0.1	$3.3 \mu\text{m/s}$	1 Hz	$31 \mu\text{m}$	3×3	Superparamagnetic microparticles
26	0.143	$500 \mu\text{m/s}$	7 Hz	$500 \mu\text{m}$	1×6	Polydimethylsiloxane
27	0.0214	$75 \mu\text{m/s}$	10 Hz	$350 \mu\text{m}$	3×3	Polydimethylsiloxane

28	0.011	9 $\mu\text{m/s}$	34 Hz	25 μm	3000	Silicone elastomer
29	0.25	83 $\mu\text{m/s}$	0.083 Hz	4 mm	8 \times 8	Silicone elastomer
30	0.38	0.95 mm/s	2.5 Hz	1 mm	6 \times 6	Silicone elastomer
This work	0.147	0.195 mm/s	1.1 Hz	1.2 mm	1\times7	Ferrofluid

Table R3.1. Comparison of the fluid pumping performance using artificial cilia with literature. The normalized velocity, shows how fast the flow it can generate by one beating cycle.

References

1. Kose, A. R., Fischer, B., Mao, L., Koser, H. Label-free cellular manipulation and sorting via biocompatible ferrofluids. *Proc. Natl. Acad. Sci. U. S. A.* **2009**, *106*, 21478-21483.
2. Arana, M., Bercoff, P. G., Jacobo, S. E., Zélis, P. M., Pasquevich, G. A. Mechanochemical synthesis of MnZn ferrite nanoparticles suitable for biocompatible ferrofluids. *Ceram. Int.* **2016**, *42*, 1545-1551.
3. Khoramian, S., Saeidifar, M., Zamanian, A., Saboury, A. A. Synthesis and characterization of biocompatible ferrofluid based on magnetite nanoparticles and its effect on immunoglobulin G as an immune protein. *J. Mol. Liq.* **2019**, *273*, 326-338.

4. Kole, M., & Khandekar, S. Engineering applications of ferrofluids: A review. *J. Magn. Magn. Mater.* **2021**, 537, 168222.
5. Socoliuc, V., Avdeev, M. V., Kuncser, V., Turcu, R., Tombácz, E., Vekas, L. Ferrofluids and bio-ferrofluids: looking back and stepping forward. *Nanoscale* **2022**, 14, 4786-4886.
6. Yang, Z., Zhang, L. Magnetic actuation systems for miniature robots: A review. *Adv. Intell. Syst.* **2020**, 2, 2000082.
7. Chesters, A. Modelling of coalescence processes in fluid-liquid dispersions: a review of current understanding. *Chem. Eng. Res. Des.* **1991**, 69, 259-270.
8. Jin, J., Ooi, C. H., Dao, D. V., Nguyen, N. T. Coalescence processes of droplets and liquid marbles. *Micromachines* **2017**, 8, 336.
9. Hassan, M. R., Wang, C. Ferro-hydrodynamic interactions between ferrofluid droplet pairs in simple shear flows. *Colloids Surf. A Physicochem. Eng. Asp.* **2020**, 602, 124906.
10. Vilfan, M., Potočnik, A., Kavčič, B., Osterman, N., Poberaj, I., Vilfan, A., Babič, D. Self-assembled artificial cilia. *Proc. Natl. Acad. Sci. U. S. A.* **2010**, 107, 1844-1847.
11. Rockenbach, A., Schnakenberg, U. The influence of flap inclination angle on fluid transport at ciliated walls. *J. Micromechanics Microengineering* **2017**, 27, 015007.
12. Zhang, S., Wang, Y., Lavrijsen, R., Onck, P. R., den Toonder, J. M. Versatile microfluidic flow generated by moulded magnetic artificial cilia. *Sensors Actuators B Chem.* **2018**, 263, 614-624.

13. Shields, A. R., Fiser, B. L., Evans, B. A., Falvo, M. R., Washburn, S., Superfine, R. Biomimetic cilia arrays generate simultaneous pumping and mixing regimes. *Proc. Natl. Acad. Sci. U. S. A.* **2010**, *107*, 15670-15675.
14. Gu, H., Boehler, Q., Cui, H., Secchi, E., Savorana, G., De Marco, C., Gervasoni, S., Peyron, Q., Huang, T., Pane, S., Hirt, A. M., Ahmed, D., Nelson, B. J. Magnetic cilia carpets with programmable metachronal waves. *Nat. Commun.* **2020**, *11*, 1-10.
15. Dong, X., Lum, G. Z., Hu, W., Zhang, R., Ren, Z., Onck, P. R., Sitti, M. Bioinspired cilia arrays with programmable nonreciprocal motion and metachronal coordination. *Sci. Adv.* **2020**, *6*, eabc9323.

List of Changes in Manuscript

1. In paragraph 1 of **Abstract**, “However, most soft machines are limited to solid magnetic materials, whereas further progress also relies on fluidic constructs obtained by reconfiguring magnetic liquid materials (e.g., ferrofluid) into soft machines.” is changed to “**However, most soft machines are limited to solid magnetic materials, whereas further progress also relies on fluidic constructs obtained by reconfiguring magnetic liquid materials (e.g., ferrofluid) into soft machines.**”.
2. In paragraph 2 of **Introduction**, “Despite many advances in using ferrofluid to construct miniature soft machines, existing machines based on ferrofluid droplets only consider the case at low wettability with the substrate (contact angle θ_c , $90^\circ \leq \theta_c < 180^\circ$), meaning that the interaction strength between solid-liquid is extremely weak⁴²⁻⁴⁶. However, there are various wetting dynamics between the ferrofluid droplet and the interface; in addition to the low wetting case, there is also high wetting and complete wetting, but the application scenarios for different wetting characteristics between ferrofluid droplets and substrates have not been fully explored. Moreover, these ferrofluid droplet-based machines are limited to a few simple modes of motion, such as stretching and rolling, due to their reliance on magnetic field gradient forces for actuation. Furthermore, while bringing better adaptability to these droplet-based machines, the high deformability limits their mechanical properties. In-depth studies have not been performed to exploit the behavior of millimeter-scale ferrofluid droplets in wetting with different interfaces and the individual dynamics under magnetic torque; thus, it is challenging to realize their full potential to be reconfigured as miniature soft machines.” is changed to “**Despite many advances in using ferrofluid to construct miniature soft machines, existing machines based on ferrofluid**

droplets only consider the case at low wettability with the substrate (contact angle θ_c , $90^\circ \leq \theta_c < 180^\circ$), meaning that the interaction strength between solid-liquid is extremely weak⁴²⁻⁴⁶. However, there are various wetting dynamics between the ferrofluid droplet and the interface⁴⁴⁻⁵⁰; in addition to the low wetting case, there is also high wetting and complete wetting, but the application scenarios for different wetting characteristics between ferrofluid droplets and substrates have not been fully explored. Moreover, these ferrofluid droplet-based machines are limited to a few simple modes of motion, such as stretching and rolling, due to their reliance on magnetic field gradient forces for actuation. Furthermore, while bringing better adaptability to these droplet-based machines, the high deformability limits their mechanical properties. The dynamic behavior of millimeter-scale ferrofluid droplets in wetting with different interfaces is underutilized, and the individual dynamics under magnetic torque have not been adequately investigated; thus, it is challenging to realize their full potential to be reconfigured as miniature soft machines.”.

3. In paragraph 3 of **Introduction**, “When the interaction between ferrofluid and substrate is weak (low wettability), the magnetic torque generated by the spatiotemporally programmed magnetic field drives ferrofluid droplets to perform stretching, jumping, rotating, tumbling, kayaking, and wobbling motions. Due to their liquid properties, the ferrofluid droplets can split and fuse along the line or the plane in a controlled manner, and the number of fissions is highly controlled. The ferrofluid droplets are highly adaptable to both structured and unstructured environments by exploiting multimodal motion and controllable fission-fusion properties. The ferrofluid droplets can cross over successive obstacles, upstairs and through designated holes in walls in a jumping mode; navigate through highly curved small gaps and sharp turns in a stretching pattern; pass through narrow comb-like channels

using fission and fusion. Moreover, the ferrofluid droplets can be reconfigured into miniature machines with multiple functions using wetting dynamics. At low wettability, the droplets can be reconfigured to serve as wireless liquid capsules for transporting liquid or solid cargo, which can travel through tortuous narrow channels to reach targeted positions and release the load-on-demand. When the interaction between ferrofluid droplets and the interface is strong (high wettability, $0^\circ < \theta_c < 90^\circ$), the controllable deformability of the droplets allows them to act as arrays of liquid cilia, which are programmed to pump biological fluids. When the interaction between ferrofluid droplets and the interface is very strong (total wetting, $\theta_c = 0^\circ$), the droplets can be reconfigured to serve as an active wireless liquid skin, which can controllably navigate near inanimate targets and then transform it into a soft machine through an adhesive strategy.” is changed to “When the interaction between ferrofluid and substrate is weak (low wettability with substrate), the magnetic torque generated by the spatiotemporally programmed magnetic field drives ferrofluid droplets to perform stretching, jumping, rotating, tumbling, kayaking, and wobbling motions. Due to their liquid properties, the ferrofluid droplets can split and fuse along the line or the plane in a controlled manner, and the number of fissions is highly controlled. Compared with the ferrofluid droplets driven by the magnetic field gradient force^{36,37,41}, the magnetic torque-driven ferrofluid droplets have various motion modes and split modes. The ferrofluid droplets are highly adaptable to both artificial and biological environments by exploiting multimodal motion and controllable fission-fusion properties. The ferrofluid droplets can cross over successive obstacles, upstairs and through designated holes in walls in a jumping mode, navigate through highly curved small gaps and sharp turns in a stretching pattern, and pass through narrow comb-like channels using fission and fusion. Moreover, the ferrofluid

droplets can be reconfigured into miniature machines with multiple functions using wetting dynamics. At low wettability with cargo, the droplets can be reconfigured to serve as liquid capsules for transporting liquid or solid cargo, which can travel through tortuous narrow channels to reach targeted positions and release the load-on-demand. However, it is hard for the solid capsule to traverse cavities when its size is comparable with the cross-sectional dimension of these confined spaces. When the interaction between ferrofluid droplets and the interface is strong (high wettability, $0^\circ < \theta_c < 90^\circ$), the controllable deformability of the droplets allows them to act as arrays of liquid cilia, which are programmed to pump biological fluids. Conventional artificial cilia are in a solid state^{18,27}, and their morphology cannot be easily changed, so they maintain a rigid structure without a magnetic field, which may affect fluid flow. Without an applied magnetic field, our liquid cilia will shrink to the bottom of the substrate and take on a spherical shape, thus reducing the impact on fluid flow. When the interaction between ferrofluid droplets and the interface is very strong (total wetting, $\theta_c = 0^\circ$), the droplets can be reconfigured to serve as an active wireless liquid skin, which can controllably navigate near inanimate targets and then transform it into a soft machine through an adhesive strategy. Unlike the passive magnetic spray²⁸, our ferrofluid droplet has multimodal motion capabilities that act as active, movable skin. When the interaction between ferrofluid and substrate is weak (low wettability with the substrate), the magnetic torque generated by the spatiotemporally programmed magnetic field drives ferrofluid droplets to perform stretching, jumping, rotating, tumbling, kayaking, and wobbling motions.”.

4. In paragraph 1 of **Wetting dynamics of ferrofluid droplets**, “The contact angle is different from hydrocarbon oil with various substrates in the air environment (**Figure S2**).” is added.

5. In paragraph 1 of **Wetting dynamics of ferrofluid droplets**, “Furthermore, with the external magnetic field, the morphology of the ferrofluid droplets on the different substrates changes as the magnetic field strength varies (0-9 mT) (**Figure 2B**).” is changed to “**Furthermore, with the external magnetic field, the morphology of the ferrofluid droplets on the different substrates changes as the magnetic field strength varies from 0 to 9 mT (Figure 2B).**”.
6. In paragraph 1 of **Wetting dynamics of ferrofluid droplets**, “**The hydrophobicity of the matrix mainly determines the wetting properties of ferrofluid droplets. Therefore, different hydrophobic substrates can be selected to construct miniature soft robots according to the needs.**” is added.
7. In the figure captions of **Figure 2. Wetting dynamics of the ferrofluid droplet** “**All the suspending phase is water.**” is added.
8. In paragraph 1 of **Individual ferrofluid droplet dynamics**, “When ferrofluid droplets do not adhere to the substrate (low wettability), changing the spatial configuration of the external magnetic field enables locomotion modalities like stretching, jumping, rotating, tumbling, kayaking, and wobbling (**Video S1**, the spatial distribution states of magnetic fields can be queried in **Figure S3**).” is changed to “**When ferrofluid droplets do not adhere to the substrate, which is made of hydrogel (low wettability), changing the spatial configuration of the external magnetic field enables locomotion modalities like stretching, jumping, rotating, tumbling, kayaking, and wobbling (Video S1, the spatial distribution states of magnetic fields can be queried in Figure S4).**”.
9. In the figure captions of **Figure 3. Fission and fusion scenarios**, “**All the substrates are made of hydrogel.**” is added.
10. In paragraph 2 of **Individual ferrofluid droplet dynamics**, “**As the frequency decreases, the magnetic dipolar force on the ferrofluid sub-droplet dominates its dynamic behavior. The ferrofluid droplets begin to approach**

each other due to the magnetic attraction. As the sub-droplets proceed into close contact, their interfaces flatten, and the fluid film between the droplets drains. Eventually, when the fluid film becomes thin enough, the van der Waals intermolecular forces dominate and cause the liquid film to rupture, ultimately contributing to droplet coalescence.” is added.

11. In paragraph 1 of **Environmental adaptability**, “It remains challenging for miniature robots to navigate freely in structured environments with obstacles, textures, and variations. To tackle this challenge, a combination of multimodal motion, deformation, and scale reconfiguration capabilities enables controlled locomotion of ferrofluid droplets in complex unstructured environments, such as continuous fences, wide gaps, uneven bladder lining, folded stomach lining, narrow lumen (**Video S3** and **Video S4**)” is changed to “It remains challenging for miniature robots to navigate freely in complex environments with obstacles, textures, and variations. To tackle this challenge, when the ferrofluid is at low wetting with the surrounding environment, a combination of multimodal motion, deformation, and scale reconfiguration capabilities enables controlled locomotion of ferrofluid droplets in complex artificial and biological environments, such as continuous fences, wide gaps, uneven bladder lining, folded stomach lining, narrow lumen (**Video S3** and **Video S4**).”.

12. In paragraph 1 of **Environmental adaptability**, “When moving inside an annular channel with an inner diameter of 2 mm, the ferrofluid droplet constantly stretches and shrinks to achieve locomotion by utilizing the frictional force of the sidewall.” is changed to “When moving inside an annular channel with an inner diameter of 2 mm, the ferrofluid droplets are constantly turning as they stretch and contract to achieve locomotion by utilizing the supporting force of the sidewall (**Figure S16**).”.

13. In paragraph 2 of **Environmental adaptability**, “In addition, in vitro cytotoxicity tests have shown that ferrofluid droplets are biocompatible (**Figure S17**).” is added.
14. In paragraph 2 of **Environmental adaptability**, “By combining multimodal motion, deformation, and scale reconfiguration capabilities, the ferrofluid droplet can navigate freely in complex environments with highly variable feature sizes, which is expected to be applied as wireless soft medical robots in various restricted areas of the human body.” is changed to “**Experimental results show that when the ferrofluid is at low wetting with the surrounding environment, by combining multimodal motion, deformation, and scale reconfiguration capabilities, the ferrofluid droplet can navigate freely in complex environments with highly variable feature sizes, which is expected to be applied as wireless soft medical robots in various restricted areas of the human body.**”.
15. In the figure captions of **Figure 4. Multimodal locomotion, deformation and scale-reconfiguration over the complex environment**, “**All the artificial structures are made of hydrogel.**” is added.
16. The title of **Miniature wireless liquid capsule at low wettability or non-wetting** is changed to “**Miniature liquid capsule at low wettability or non-wetting**”.
17. In paragraph 1 of **Miniature wireless liquid capsule at low wettability or non-wetting**, “**But the injection of liquid cargo requires complete wetting between the syringe needle and the ferrofluid droplet (**Figure S18**). To avoid instability and ejection, a typical 5 μL droplet is injected with a liquid load of approximately 0.5 μL .**” is added.
18. In paragraph 1 of **Miniature wireless liquid capsule at low wettability or non-wetting**, “**The mechanisms of drug release and mechanisms to maintain non-leakage during motion is shown in **Figure S19**.**” is added.

19. In the figure captions of **Figure 5. Controllable liquid capsule for on-demand cargo delivery**, “(D) Liquid capsules maneuver in various motion modes while carrying liquid or solid cargo and release on demand. (i) Liquid cargo delivery. (ii) Solid cargo delivery and mixing. (iii) Two liquid capsules are actuated together to deliver the liquid cargo and then fuses. (E) The liquid capsule shuttles back and forth in the channel for 40 minutes before ejecting the liquid cargo. (F) The liquid capsule moves in a bile duct phantom with narrow branches in stretching mode and release the liquid cargo on demand.” is changed to “(D) Liquid capsules maneuver in various motion modes while carrying the liquid or solid cargo and release on demand. (i) Liquid cargo delivery. (ii) Solid cargo delivery and mixing. (iii) Two liquid capsules are actuated together to deliver the liquid cargo and fuse. The substrate is made of hydrogel. (E) The liquid capsule shuttles back and forth in the channel, which is made of hydrogel, for 40 minutes before ejecting the liquid cargo. (F) The liquid capsule moves in a bile duct phantom made of soft gel, inside whose narrow branches are in stretching mode and release the liquid cargo on demand.”.

20. In paragraph 1 of **Miniature wireless liquid cilia at high wettability**, “According to the scallop theorem, this nonreciprocal motion is essential for the cilia to pump the liquid in their vicinity at low Reynolds numbers²⁷. Unlike natural or other artificial cilia, this presented liquid cilia based on ferrofluid exhibit nonreciprocal motion, even though they do not have a "power stroke" or a "recovery stroke." In its elongation-rotation-contraction motion cycle, the area where the rotational motion travels are the swiping area (the swiping area is a common metric for quantifying nonreciprocal motion and the net fluid flow-induced). As the swiping area increases, the motion of the liquid cilia becomes more nonreciprocal, resulting in increased fluid flow at low Reynolds numbers. Furthermore, the size of the swiping

area of the liquid cilia can be controlled by programming the amplitude of the external magnetic field and the angle of rotation.” is changed to “**Unlike natural cilia or other artificial cilia made from elastomer depending on bucking motion²⁷, this presented liquid cilia based on elongation and contraction exhibit nonreciprocal motion with periodic power stroke and recovery stroke. In its elongation-rotation-contraction motion cycle, the area where the rotational motion travels is the swiping area (the swiping area is a common metric for quantifying nonreciprocal motion and the net fluid flow-induced). As the swiping area increases, the motion of the liquid cilia becomes more nonreciprocal, resulting in increased fluid flow . Furthermore, the size of the swiping area of the liquid cilia can be controlled by programming the amplitude of the external magnetic field and the angle of rotation.**”.

21. In paragraph 2 of **Miniature wireless liquid cilia at high wettability**, “**The fluid velocity is related to the periodicity and beat frequency of the array is shown in the **Figure S26**. And the performance of liquid cilia compared to other artificial cilia at low Reynolds (Re) numbers is shown in **Table S1**.**” is added.
22. In the figure captions of **Figure 6. Programmable liquid cilia as fluidic devices**, “**The substrate of the liquid cilia is a transparent resin material.**” is added.
23. In paragraph 1 of **Miniature wireless liquid skin at total wetting**, “In the first stage, ferrofluid penetrates the silicone elastomer surface layer. Due to the high mutual solubility of the hydrocarbon solvent and the elastomeric sheet, the solvent diffuses rapidly from the surface to the inner layer along its concentration gradient. As a result, the upper part of the elastomer sheet becomes swollen and inflated. At the same time, most of the dispersed nanoscale iron oxide particles concentrate and precipitate out, and some of

them adhere to the surface. In the second stage, ferrofluid penetrates the body of the elastomer sheet. The solvent continues to evaporate, and the magnetic particles start to agglomerate. In the third stage, the ferrofluid completely penetrates the body of the elastomer sheet. Iron oxide nanoparticles aggregate on the scanned surface and form a continuous rigid layer.” is changed to “In the first stage, the ferrofluid droplet is driven by the magnetic field to reach the target elastomer and then, under the influence of the magnetic field, thoroughly wets the elastomer surface. In the second stage, ferrofluid penetrates the silicone elastomer surface layer. Due to the high mutual solubility of the hydrocarbon solvent and the elastomeric sheet, the solvent diffuses rapidly from the surface to the inner layer along its concentration gradient. As a result, the upper part of the elastomer sheet becomes swollen and inflated. At the same time, most dispersed nanoscale iron oxide particles concentrate and precipitate out, and some adhere to the surface. In the third stage, ferrofluid penetrates the body of the elastomer sheet. The solvent continues to evaporate, and the magnetic particles start to agglomerate. Iron oxide nanoparticles aggregate on the scanned surface and form a continuous rigid layer.”.

24. In paragraph 1 of **Miniature wireless liquid skin at total wetting**, “The propulsion mechanism for the fish robot is the same as the body-caudal fin propulsion swimming robot^{51,52}. The head of the fish robot is firstly wetted by the ferrofluid droplet, which enables it to be controlled by the external magnetic field. The oscillating magnetic field will force the robot's head to sway from side to side, a process similar to excitation at one terminal of a beam at its first-order bending vibration frequency, thus causing the beam to bend. Since the head is heavier and more rigid (since the elastomer sheet at this region is wetted by the ferrofluid droplet), the tail displacement will be more pronounced. Then when the tail swings from side to side, the water on

the rear side is continuously pushed away from the body, causing vortices behind its tail as the COMSOL Multiphysics simulation result in **Figure S32**, and the reaction force generated by the fluid on the robot's body during this process will eventually push the robot forward.” is added.

25. In the figure captions of **Figure 7. Mobile liquid skin converts inanimate objects to miniature soft machines**, “All the substrates are made of hydrogel.” is added.

26. In paragraph 1 of **Conclusion and discussion**, “Our results show that ferrofluid droplets' reconfigurability and wetting properties can be programmed to construct multifunctional miniature soft machines to address environmental changes and multitasking requirements.” is changed to “Our results show that ferrofluid droplets' reconfigurability and wetting properties can be integrated to construct multifunctional miniature soft machines to address environmental changes and multitasking requirements.”.

27. In paragraph 2 of **Conclusion and discussion**, “Moreover, the wetting properties of other solid interfaces can be programmed to change, and future work can focus on altering the wettability of substrates to achieve mobile liquid skin attachment and desorption.” is added.

28. In paragraph 1 of **Materials and methods**, “The placentas of the newborn babies used in the experiments were collected from the Prince of Wales Hospital, Hong Kong. The collection of human placentas was approved and overseen by The Joint Chinese University of Hong Kong-New Territories East Cluster Clinical Research Ethics Committee (The Joint CUHK-NTEC CREC) (Ref. No. 2020.384). All enrolled patients provided written informed consent. The human placenta was donated by pregnant women in collaboration with the Department of Obstetrics and Gynaecology (CUHK).” is added.

29. In paragraph 2 of **Materials and methods**, “Here, fresh bile ducts, bladders and stomachs of pigs were purchased from the market, dissected and rinsed with water.” is added.

30. New references, including [39,49-52], are added.

List of Changes in Supporting Information

1. New Figures S2, S16, S17, S18, S19, S26, S32 are added.
2. New Table S1 is added.

REVIEWER COMMENTS

Reviewer #1 (Remarks to the Author):

In this round of revision, the authors have addressed some of my comments. Here are some following-up comments to the author's responses.

1. I am surprised that the authors deleted my comment "The novelty of this work is very limited." when sending back their response. I am still not convinced that the novelty of this work is sufficient. First, Fig. 2 to Fig. 5 shows non-wetting of the ferrofluid droplet on hydrogel surfaces, which have been shown in many works (Fan et al. PNAS 2020). Second, For Fig. 6, I am still concerned with the advantage of the using the ferrofluid as artificial cilia over works in literature. The step-out frequency is very small - less than 1 Hz and the pumping speed at low $Re\#$ is worse than several works due to the softness of the droplets. The device is also difficult to be scaled down as the surface tension would be more dominant at small scales making it difficult to elongate the ferrofluid droplet at smaller scales. Lastly, the robots in Fig. 7 are slow, which didn't show any advantage over existing elastomer-based magnetic soft robots in literature.

2. The authors tried to avoid "programming" but the title "Reconfigurable ferrofluidic wetting for miniature soft machines" is still very misleading. As the authors agreed, they cannot change the surface wetting properties currently. So, the wetting property of the ferrofluid on a given substrate is fixed in this work. How do you reconfigure the wetting?

3. The contact angle of the ferrofluid and hydrocarbon oil on hydrogel is around 10 degrees in Figure R1.2, which is contradictory to the non-wetting conditions used in Fig. 2 to Fig. 5.

4. "Our results show that ferrofluid droplets' reconfigurability and wetting properties can be integrated to construct multifunctional miniature soft machines to address environmental changes and multitasking requirements." This is largely an overclaim as no demonstrations have been shown for addressing the environmental changes and proving the multiple task ability beyond existing works.

5. The pull off force in Fig. 2E should depend on the magnetic field spatial gradient rather than the magnetic field strength itself as magnetic field only induces torque rather than force.

6. In Fig. S32, are the simulation a two-way coupling modeling? There is little detail about the simulation and modeling. Please include more details.

Reviewer #2 (Remarks to the Author):

In their resubmittal letter, the authors have addressed in detail all points of my previous report. They have modified their manuscript accordingly. Thus, I support the publication of the manuscript in its present form in Nature Communications.

Reviewer #3 (Remarks to the Author):

The authors have addressed my comments. I can recommend this work for publication.

Response to Reviewer #1

In this round of revision, the authors have addressed some of my comments. Here are some following-up comments to the author's responses.

Response: We thank the reviewer for their careful review. Based on the reviewer's comments, we have substantially revised our manuscript by clarifying the contribution, deleting the overstated claims and reorganizing/rewriting the paper. Please check the following point-by-point response for details.

1. I am surprised that the authors deleted my comment "The novelty of this work is very limited." when sending back their response. I am still not convinced that the novelty of this work is sufficient. First, Fig. 2 to Fig. 5 shows non-wetting of the ferrofluid droplet on hydrogel surfaces, which have been shown in many works (Fan et al. PNAS 2020). Second, For Fig. 6, I am still concerned with the advantage of the using the ferrofluid as artificial cilia over works in literature. The step-out frequency is very small - less than 1 Hz and the pumping speed at low $Re\#$ is worse than several works due to the softness of the droplets. The device is also difficult to be scaled down as the surface tension would be more dominant at small scales making it difficult to elongate the ferrofluid droplet at smaller scales. Lastly, the robots in Fig. 7 are slow, which didn't show any advantage over existing elastomer-based magnetic soft robots in literature.

Response: We thank the reviewer for the comments. We sincerely apologize for missing this comment in the copy-and-paste process. In our previous reply, we made an attempt to respond specifically to the reviewer's concerns about limited

novelty. Here, we try to further elucidate the differences and uniqueness of our work in the dedicated sections below.

(i) Regarding the non-wetting of the ferrofluid droplet on hydrogel surfaces, we agree with the reviewer that this has been shown in previous works^[1,2]. We have made sure to highlight the contributions of these previous studies in the manuscript. However, these ferrofluid droplet-based machines on non-wetting hydrogel surfaces are limited to a few simple modes of motion, such as stretching and rolling, due to their reliance on magnetic field gradient forces for actuation. In contrast, here we show the ability to use magnetic torques generated by the spatiotemporally programmed magnetic field to drive ferrofluid droplets to perform a broader range of motions—stretching, jumping, rotating, tumbling, kayaking, and wobbling. Due to their liquid properties, the ferrofluid droplets can split or fuse in a controlled manner, and the number of fusing droplets can be highly controlled. To summarize, compared with the ferrofluid droplets driven by the magnetic field gradient force, the magnetic torque-driven ferrofluid droplets have various modes of motion and splitting/coalescence. A discussion of comparison with previous work has been added to the revised manuscript (**please see section 3 and page 26**).

(ii) Regarding the advantage of using the ferrofluid as artificial cilia, we show that using ferrofluid droplet enables the length of artificial cilia to be dynamically adjusted. This is determined by the strength of the external magnetic field and is in contrast to artificial cilia based on magnetic elastomers, which have fixed dimensions^[3,4]. We know that the farther away from the cilia position, the smaller the fluid velocity. It isn't easy to reconfigure a cilium with a fixed length to meet the fluid pumping farther away from the cilia. The adjustable length of the liquid cilia allows the liquid cilia to be reconfigured *in situ* for more flexible fluid pumping. As shown in **Figure R1.1a**, the size of the

static liquid cilia can be elongated to 5 times the initial length (1 mm) under the magnetic field of 18 mT. In addition, the step-out frequency of liquid cilia based on ferrofluid droplets is determined by the magnetic strength. The cut-off frequency is 1.1 Hz when the magnetic field strength is 9 mT. When the magnetic field strength increases to 18 mT, the cut-off frequency is about 4.2 Hz (**Figure R1.1b**).

Moreover, the pumping speed of liquid cilia is not determined by the frequency alone. We can also increase the pumping speed by adjusting the external magnetic field to increase the cilia length and oscillation angle. For example, in the previous response letter, we pointed out that when the magnetic field strength is 9 mT, the cut-off frequency of the cilia array is 1.1 Hz, the length of dynamic cilia is 1.2 mm, and when its oscillation angle is set as 120° , its pumping fluid velocity is about 0.195 mm/s. However, when the magnetic field strength is increased to 18 mT, the cut-off frequency of the cilia array is 4.2 Hz, the length of dynamic cilia is about 1.5 mm, and when the oscillation angle is increased to 150° , and finally, the pumping velocity will increase to 0.97 mm/s. This pumping speed at a low Reynolds number is not worse than previous works^[3,4], and the pumping rate of cilia can be further increased by increasing the strength of the external uniform magnetic field.

Furthermore, our experimental results show that the liquid cilia can be scaled down to 500 microns for pumping (**Figure R1.1c**) and that 500 μm cilia can elongate to 750 μm under a 15 mT magnetic field. We agree with the reviewer that if the liquid cilia are further scaled to smaller sizes, the surface tension of the liquid cilia will dominate. At this point, limitations on the magnetic field strength generated by the Helmholtz electromagnetic coil setup make it difficult to deform. We can use devices that create larger magnetic field strengths to drive smaller-scale liquid cilia, such as the Halbach array^[4,7]. Alternatively, the

surface tension of ferrofluid droplets can be changed by changing surfactants to prepare smaller liquid cilia^[8]. These results have been added to the revised manuscript (please see section 2.5 and pages 20 and 21).

Figure R1.1. The performance of liquid cilia. **a.** The relationship between liquid cilia length and magnetic field strength. **b.** Cut-off frequency of liquid cilia array (1×7) versus magnetic field strength (rotation angle of 120°). **c.** Snapshots of the liquid cilia scaled to $500 \mu\text{m}$. Scale bar: $500 \mu\text{m}$.

(iii) Regarding the magnetic coating, our ferrofluid droplet has multimodal motion capabilities that act as active, movable “skin”. We agree with the reviewer that these ferrofluid-based robots are slower than the existing elastomer-based magnetic soft robots. However, we focus on how this liquid skin can use its adhesive properties to enable inanimate objects to move by coating them with a thin, magnetically drivable film. This is in contrast to manipulation tasks with other magnetic soft robots, in which the locomotion speed of the robot limits the motion of the object. In addition, this ferrofluid-based preparation strategy allows *in situ* selection of the site of adhesion, making a target part of an inanimate object capable of movement. The existing magnetic elastomer-based soft robots are difficult to reconfigure *in situ* once the

processing is completed. The advantage of this movable skin is that it can enter some narrow spaces and then use its adhesive properties to endow inanimate objects with the ability to move, thus allowing remote manipulation of the target object.

In short, compared with the ferrofluid droplets driven by the magnetic field gradient force, the magnetic torque-driven ferrofluid droplets have a wider variety of locomotion and deformation modes. Moreover, liquid cilia based on ferrofluid can exhibit dynamically adjustable length and rotation angle. Its maximum cut-off frequency is about 4.2 Hz when the magnetic field strength is 18 mT, and the current maximum pumping speed is about 0.97 mm/s. Lastly, the droplets can be reconfigured to serve as an active liquid skin that can be controllably positioned near an inanimate target and then bond to the target to transform it into a magnetically-responsive soft machine. This dynamic and movable liquid skin is capable of selective adhesion and objects manipulation or locomotion.

2. The authors tried to avoid “programming” but the title “Reconfigurable ferrofluidic wetting for miniature soft machines” is still very misleading. As the authors agreed, they cannot change the surface wetting properties currently. So, the wetting property of the ferrofluid on a given substrate is fixed in this work. How do you reconfigure the wetting?

Response: We thank the reviewer for the comments. We previously used the term reconfigurable to emphasize the ferrofluid droplets’ various modes of motion and deformability and its split-fusion behavior. According to your suggestion, to avoid being misunderstanding, the title “Reconfigurable ferrofluidic wetting for miniature soft machines” has been modified in the

revised manuscript as follows: “Exploiting ferrofluidic wetting for miniature soft machines”.

3. The contact angle of the ferrofluid and hydrocarbon oil on hydrogel is around 10 degrees in Figure R1.2, which is contradictory to the non-wetting conditions used in Fig. 2 to Fig. 5.

Response: We thank the reviewer for the comments. The contact angles of ferrofluid and hydrocarbon oil on hydrogels contradict the non-wetting conditions used in Fig. 2 to Fig. 5 because these contact angles were measured inside air and not in the water, where the hydrogel surface has been dried. This was stated in our previous response and is copied below for your reference, although we still regret the confusion and hope that the distinction is clearer now:

“Since hydrocarbon oil is less dense than water, measuring the contact angle between the hydrocarbon oil and various interfaces in the underwater environment is hard. So here, we measure and compare the contact angles of hydrocarbon oil and ferrofluid with multiple interfaces in the air environment.”

4. “Our results show that ferrofluid droplets’ reconfigurability and wetting properties can be integrated to construct multifunctional miniature soft machines to address environmental changes and multitasking requirements.” This is largely an overclaim as no demonstrations have been shown for addressing the environmental changes and proving the multiple task ability beyond existing works.

Response: We thank the reviewer for the comments. According to your suggestion, to avoid being misunderstanding, the description “Our results show that ferrofluid droplets’ reconfigurability and wetting properties can be integrated to construct multifunctional miniature soft machines to address environmental changes and multitasking requirements.” has been modified in the revised manuscript as follows: “Our results show that ferrofluid droplets’ reconfigurability and wetting properties can be integrated to construct multifunctional miniature soft machines.” (Please see section 3 and page 25).

5. The pull off force in Fig. 2E should depend on the magnetic field spatial gradient rather than the magnetic field strength itself as magnetic field only induces torque rather than force.

Response: We thank the reviewer for the comments. In the revised manuscript, the magnetic field strength has been converted into the magnetic field spatial gradient according to the measured distance data (Figure R1.2). Please see page 6 and Figure 2E of the revised manuscript for more details.

Figure R1.2. The quantitative pulling off strength of ferrofluid droplets to

different surfaces.

6. In Fig. S32, are the simulation a two-way coupling modeling? There is little detail about the simulation and modeling. Please include more details.

Response: We thank the reviewer for the comments. The simulation is a two-way coupling modeling. According to your suggestion, we have added more details of the simulation in the figure captions of **Figure S32**, which is also provided here for your reference.

“The propulsion mechanism for the fish robot is the same as the body-caudal fin propulsion swimming robot^[5,6]. The head of the fish robot is firstly wetted by the ferrofluid droplet, which enables it to be controlled by the external magnetic field. The oscillating magnetic field will force the robot's head to sway from side to side, a process similar to excitation at one terminal of a beam at its first-order bending vibration frequency, thus causing the beam to bend. Since the head is heavier and more rigid (since the elastomer sheet at this region is wetted by the ferrofluid droplet), the tail displacement will be more pronounced. Then when the tail swings from side to side, the water on the rear side is continuously pushed away from the body, causing vortices behind its tail (as the COMSOL Multiphysics simulation shows). The reaction force generated by the fluid on the robot's body during this process will eventually push the robot forward. For the simulation, the Fluid-Structure Interaction module is applied, the length of the magnetic head of the fish robot is set to 5 mm and the thickness is 1.5 mm, and the length of the non-magnetic tail is set to 23 mm and the thickness is 1 mm. The oscillation frequency of the head of the fish robot is 4 Hz. The element size parameters include the maximum element size of 0.983 mm, the minimum element size of 0.0044 mm, the maximum element growth rate of 1.3, the

curvature factor of 0.3, and the resolution of narrow regions of 1. The relationship between the deformation of the fish robot body and the fluid force is given by

$$\rho_s \ddot{\mathbf{u}}_s = \text{div } \mathbf{S} + \mathbf{f}_v \quad (1)$$

where ρ_s is the mass density of the fish robot, \mathbf{u}_s is the displacement vector, \mathbf{S} is the reference stress applied to the fish robot, \mathbf{f}_v is the fluid force exerted on the solid structure. The fluid is governed by the forces balance and mass conservation equations as follows:

$$\rho_f \dot{\mathbf{v}}_f + \rho_f (\nabla \mathbf{v}_f) \mathbf{v}_f = \text{div } \Gamma + \mathbf{f} \quad (2)$$

$$\dot{\rho}_f + \text{div} (\rho_f \mathbf{v}_f) = 0 \quad (3)$$

Where ρ_f is the mass density of the fluid, \mathbf{v}_f is the fluid spatial velocity, \mathbf{f} is the force of solid acting on the fluid, and the stress Γ is given by:

$$\Gamma = -p\mathbf{I} + 2\mu_f (\text{sym } \nabla \mathbf{v}_f) - \frac{2}{3}\mu_f (\text{div } \mathbf{v}_f)\mathbf{I} \quad (4)$$

where p is the fluid pressure, and μ_f is the dynamic viscosity”.

References

1. Fan, X., Dong, X., Karacakol, A., Xie, H., Sitti, M. Reconfigurable multifunctional ferrofluid droplet robots. *Proc. Natl. Acad. Sci. U. S. A.* **2020**, *117*, 27916-27926.
2. Yu, W., Lin, H., Wang, Y., He, X., Chen, N., Sun, K., Lo, D., Cheng, B., Yeung, C., Tan, J., Carlo, D., Emaminejad, S. A ferrobatic system for automated microfluidic logistics. *Sci. Robot.* **2020**, *5*, eaba4411.

3. Gu, H., Boehler, Q., Cui, H., Secchi, E., Savorana, G., De Marco, C., Gervasoni, S., Peyron, Q., Huang, T., Pane, S., Hirt, A. M., Ahmed, D., Nelson, B. J. Magnetic cilia carpets with programmable metachronal waves. *Nat. Commun.* **2020**, *11*, 1-10.
4. Dong, X., Lum, G. Z., Hu, W., Zhang, R., Ren, Z., Onck, P. R., Sitti, M. Bioinspired cilia arrays with programmable nonreciprocal motion and metachronal coordination. *Sci. Adv.* **2020**, *6*, eabc9323.
5. Lee, K. Y., Park, S. J., Matthews, D. G., Kim, S. L., Marquez, C. A., Zimmerman, J. F., Ardoña, H. A. M., Kleber, A. G., Lauder, G. V., Parker, K. K. An autonomously swimming biohybrid fish designed with human cardiac biophysics. *Science* **2022**, *375*, 639-647.
6. Borazjani, I., Sotiropoulos, F. On the role of form and kinematics on the hydrodynamics of self-propelled body/caudal fin swimming. *J. Exp. Biol.* **2010**, *213*, 89-107.
7. Serwane, F., Mongera, A., Rowghanian, P., Kealhofer, D. A., Lucio, A. A., Hockenbery, Z. M., Campas, O. In vivo quantification of spatially varying mechanical properties in developing tissues. *Nat. Methods* **2017**, *14*, 181-186.
8. Latikka, M., Backholm, M., Timonen, J. V., Ras, R. H. Wetting of ferrofluids: Phenomena and control. *Curr. Opin. Colloid Interface Sci.* **2018**, *36*, 118-129.

List of Changes in Manuscript

1. In **Title**, “Reconfigurable ferrofluidic wetting for miniature soft machines” is changed to “**Exploiting ferrofluidic wetting for miniature soft machines**”.
2. In paragraph 2 of **Miniature wireless liquid cilia at high wettability**, “**The strength of the magnetic field determines the length of the liquid cilia, the size of the static liquid cilia can be elongated to 5 times the initial length (1 mm) under the magnetic field of 18 mT (Figure S27a). In addition, the step-out frequency of liquid cilia based on ferrofluid droplets is determined by the magnetic strength. The cut-off frequency is 1.1 Hz when the magnetic field strength is 9 mT; when the magnetic field strength increases to 18 mT, the cut-off frequency is about 4.2 Hz (Figure S27b). And the pumping speed of liquid cilia is not determined by the frequency alone. We can also increase the pumping speed by adjusting the external magnetic field to increase the cilia length and oscillation angle.**” is added.
3. In paragraph 2 of **Miniature wireless liquid cilia at high wettability**, “**The advantage of using the ferrofluid as artificial cilia is the length of artificial cilia based on ferrofluid droplets can be adjusted compared to artificial cilia based on magnetic elastomers. The adjustable length of the liquid cilia allows the liquid cilia to be reconfigured *in situ* for more flexible fluid pumping. In addition, the performance of liquid cilia compared to other artificial cilia at low Reynolds (*Re*) numbers is shown in Table S1.**” is added.
4. In paragraph 1 of **Conclusions**, “Our results show that ferrofluid droplets' reconfigurability and wetting properties can be integrated to construct multifunctional miniature soft machines to address environmental changes and multitasking requirements. Ferrofluid droplet machines can be stimulated using programmed alternating magnetic fields in various motion modes: stretching, jumping, spinning, tumbling, kayaking, and oscillating. Compared to existing soft devices based on magnetically driven elastomers,

our approach allows for more significant deformation (e.g., controlled splitting and fusion) in these droplet machines and enables complex shape deformation behavior on-demand. Jumping over high obstacles, crossing narrow channels, and maneuvering over changing textured surfaces demonstrate the high environmental adaptability of droplet machines. When coupled with different surfaces to form assembled machines, they can also assume multiple roles under different task requirements, including acting as liquid capsules for solid or liquid cargo transport, two-dimensional cilia array matrix for pumping and agitating complex biological fluids, and intelligent skin for transforming inanimate objects into miniature soft machines.” is changed to “Our results show that ferrofluid droplets' reconfigurability and wetting properties can be integrated to construct multifunctional miniature soft machines. Ferrofluid droplet machines can be stimulated using programmed alternating magnetic fields in various motion modes: stretching, jumping, spinning, tumbling, kayaking, and oscillating. Compared to existing soft devices based on magnetically driven elastomers, our approach allows for more significant deformation (e.g., controlled splitting and fusion) in these droplet machines and enables complex shape deformation behavior on-demand. Jumping over high obstacles, crossing narrow channels, and maneuvering over changing textured surfaces demonstrate the high environmental adaptability of droplet machines. Compared with the ferrofluid droplets driven by the magnetic field gradient force³⁶, the magnetic torque-driven ferrofluid droplets have various motion modes and split modes. When coupled with different surfaces to form assembled machines, they can also assume multiple roles under different task requirements, including acting as liquid capsules for solid or liquid cargo transport, two-dimensional cilia array matrix for pumping and agitating complex biological fluids, and intelligent skin for transforming

inanimate objects into miniature soft machines. The external magnetic field can determine liquid cilia's length, step-out frequency, and rotation angle. If the Halbach array device with stronger magnetic field strength is used, the pumping speed of the liquid cilia can be further increased and drives the liquid cilia of the micrometer scale. In addition, the active wireless liquid skin can controllably navigate near inanimate targets and then transform into a soft machine through an adhesive strategy. This dynamic and movable liquid skin has the advantage of selective adhesion.”.

List of Changes in Supporting Information

1. New Figure S27 is added.
2. New caption of Figure S32 is added.
3. Table S1 is modified.

REVIEWERS' COMMENTS

Reviewer #1 (Remarks to the Author):

The authors have addressed all my comments. There are no further concerns.

Response to Reviewer #1

The authors have addressed all my comments. There are no further concerns.

Response: We sincerely thank the reviewer for the kind comments and helping us improve the quality.